# Ice crystal number concentration estimates from lidar-radar satellite remote sensing. Part 2: Controls on the ice crystal number concentration

Edward Gryspeerdt[1], Odran Sourdeval[2], Johannes Quaas[2], Julien Delanoë[3], Martina Krämer[4], and Philipp Kühne[2]

[1]Space and Atmospheric Physics Group, Imperial College London, London, United Kingdom
[2]Institute for Meteorology, Universität Leipzig, Germany
[3]Laboratoire Atmosphères, Milieux, Observations Spatiales/IPSL/UVSQ/CNRS/UPMC, Guyancourt, France
[4]Forschungszentrum Jülich, Institut für Energie und Klimaforschung (IEK-7), Jülich, Germany

**Correspondence:** Edward Gryspeerdt
(e.gryspeerdt@imperial.ac.uk)

**Abstract.** The ice crystal number concentration ($N_i$) is a key property of ice clouds, both radiatively and microphysically. However, due to sparse in-situ measurements of ice cloud properties, the controls on the $N_i$ have remained difficult to determine. As more advanced treatments of ice clouds are included in global models, it is becoming increasingly necessary to develop strong observational constraints on the processes involved.

This work uses the DARDAR-LIM $N_i$ retrieval described in part one to investigate the controls of the $N_i$ at a global scale. The retrieved clouds are separated by type. The effects of temperature, proxies for in-cloud updraught and aerosol concentrations are investigated. Variations in the cloud top $N_i$ ($N_{i(top)}$) consistent with both homogeneous and heterogeneous nucleation are observed and along with differing relationships between aerosol and $N_{i(top)}$ depending on the prevailing meteorological situation and aerosol type. Away from the cloud top, the $N_i$ displays a different sensitivity to these controlling factors, providing a possible explanation to the low $N_i$ sensitivity to temperature and INP observed in previous in-situ studies.

This satellite dataset provides a new way of investigating the response of cloud properties to meteorological and aerosol controls. The results presented in this work increase our confidence in the retrieved $N_i$ and will form the basis for further study into the processes influencing ice and mixed phase clouds.

## 1 Introduction

Clouds play a central role in the Earth's energy budget, being responsible for large variations in the reflected shortwave and emitted longwave radiation (Stephens et al., 2012). The response of clouds to changing greenhouse gases and aerosols remains one of the largest uncertainties in understanding past and future climate changes (Boucher et al., 2013). Significant advances have been made into modelling and observing the role of aerosols in liquid clouds (e.g. Wang et al., 2011; Wood et al., 2011; Seifert et al., 2015; Gettelman, 2015; Ghan et al., 2016; Zuidema et al., 2016), especially through the use of retrievals of the cloud droplet number concentration (e.g. Quaas et al., 2008; Gryspeerdt et al., 2016), but the impact of aerosols on high

clouds remains uncertain (Gettelman et al., 2012; Jensen et al., 2016; Zhou et al., 2016; Heyn et al., 2017). A large part of this uncertainty comes from the difficulty in retrieving cirrus cloud properties at a large enough scale to separate the roles of individual factors controlling the ice crystal number concentration ($N_i$).

A key microphysical property of ice clouds, the $N_i$ links the aerosol environment to dynamic effects driving cloud updraughts and the generation of supersaturation (Pruppacher and Klett, 1997). Through changes in the ice crystal size, changes in the $N_i$ can have far-reaching implications for a cloud, impacting the radiative (Liou, 1986; Fusina et al., 2007), precipitation and cloud lifetime properties (Lindsey and Fromm, 2008). The $N_i$ is often used as a prognostic variable in two moment cloud microphysics schemes (e.g. Lohmann et al., 2007; Salzmann et al., 2010). This highlights a requirement to understand the controls on the $N_i$ for improving our understanding and parametrisation of cloud processes. While aircraft measurements of the $N_i$ exist, they are restricted in space and time. They can be affected by shattering of ice crystals at the instrument inlet (McFarquhar et al., 2007; Jensen et al., 2009; Korolev et al., 2013) and difficulties in measuring the smallest crystals (O'Shea et al., 2016). In this paper, the new DARDAR-LIM satellite dataset described in part one (Sourdeval et al., submitted) allows the processes that control the $N_i$ to be investigated globally.

It is known that the temperature plays a strong role in determining the ice crystal nucleation rate. The homogeneous nucleation rate is a strong function of temperature and supersaturation (Koop et al., 2000), with atmospherically relevant nucleation only taking place at temperatures colder than 235 K. This strong temperature dependence in the nucleation rate does not necessarily correspond to a strong temperature dependence in the $N_i$ (Heymsfield and Miloshevich, 1993). A weak $N_i$ temperature dependence was found by Gayet et al. (2004). Krämer et al. (2009) found similar results, with a slight reduction in the $N_i$ for the coldest measurements. Higher $N_i$ values have been observed at cold temperatures during ATTREX (Jensen et al., 2013a, 2016) than in Krämer et al. (2009), leading to a weak combined temperature dependence. However, using different datasets targeting different cloud types, Muhlbauer et al. (2014) and Jensen et al. (2013b) both showed an increase in $N_i$ with decreasing temperature, demonstrating that there is still considerable uncertainty in the $N_i$ temperature dependence.

The in-situ homogeneous nucleation of ice crystals is also dependent on the supersaturation (Koop et al., 2000; Lohmann and Kärcher, 2002), which is often generated through cooling due to vertical air motion. Large scale updraughts cannot reproduce observed cirrus properties on their own, the smaller scale variation in updraught provided by gravity waves is necessary (Kärcher and Ström, 2003) and is occasionally able to produce cirrus in regions of large scale subsidence (Muhlbauer et al., 2014). These small scale updraughts can produce $N_i$ values as high as $50{,}000\,\mathrm{L}^{-1}$ (Hoyle et al., 2005), highlighting the important role that vertical motion can play in determining the $N_i$.

Although ice can form by in-situ nucleation, many ice crystals also form through the freezing of liquid condensate. This liquid-origin cirrus often originates from high updraught regions in mixed-phase clouds, forming thicker cirrus that those composed of in-situ ice (Krämer et al., 2016; Luebke et al., 2016). Synoptic-scale updraughts can also produce liquid-origin cirrus in the mid-latitudes (Wernli et al., 2016). The $N_i$ formed through these liquid-origin processes is also strongly dependent on the cloud-scale updraughts, with higher updraughts maintaining higher ice supersaturations and producing larger $N_i$ values (Kärcher and Seifert, 2016; Kärcher, 2017).

Aerosol also plays a role in determining the $N_i$, although its impact is complicated by variations in ice crystal nucleation pathways and aerosol properties. While any particle can theoretically act as a homogeneous nucleation centre given a high enough supersaturation, in practice these aerosols are often hydrophilic liquid aerosols (Kojima et al., 2004). Increases in the aerosol number can result in an increase in $N_i$ through increased homogeneous nucleation. However, in many situations, the $N_i$ is limited by dynamical concerns, limiting the impact of aerosols (Demott et al., 1997; Lohmann and Kärcher, 2002; Kay and Wood, 2008; Barahona and Nenes, 2008; Jensen et al., 2013a, 2016).

A second class of aerosols, known as ice nucleating particles (INP) are able to nucleate ice heterogeneously and can freeze liquid water droplets at temperatures warmer than 235 K. At these warmer temperatures, the presence of INP will often control the $N_i$ near nucleation locations (Kärcher and Lohmann, 2003), as they form the sites for creating an ice crystal, although the nucleating ability of these INP is also a strong function of temperature (Hoose and Möhler, 2012). As heterogeneous nucleation can take place at a lower supersaturation than homogeneous nucleation, the introduction of INP has the ability to prevent homogeneous nucleation by depressing the supersaturation. As the $N_i$ produced through homogeneous nucleation events is typically higher than the INP concentration (and so the $N_i$ from heterogeneous nucleation), this implies that the introduction of INP into a clean atmosphere can reduce the $N_i$ (Demott et al., 1997; Kärcher and Lohmann, 2003). In-situ (Gayet et al., 2004) and satellite studies (Chylek et al., 2006) have provided some evidence for a possible decrease in $N_i$ with increasing aerosol based on regional and hemispheric differences in ice crystal properties, although has proved difficult to conclusively link these $N_i$ changes to a change in INP.

The relative role of heterogeneous and homogeneous nucleation in the atmosphere is unclear, making it difficult to develop observational constraints on the impact of aerosols on the $N_i$ (e.g. Cziczo et al., 2013; Gasparini and Lohmann, 2016; Kärcher and Seifert, 2016; Jensen et al., 2016). In addition, changing conditions over the lifecycle of a cloud can result in a switch between nucleation mechanisms (Krämer et al., 2016) and nucleation is not the only control on the $N_i$. The rarity of INP suggests that other processes, such as ice multiplication, are required to explain the $N_i$ observed in the lower atmosphere (Heymsfield et al., 2017).

These four factors (temperature, supersaturation/updraught, ice origin and aerosol environment) are all thought to influence the $N_i$ in high clouds, but there remain significant uncertainties in assessing the role of these factors on the $N_i$. First, although they have been investigated using aircraft and balloon measurements (Podglajen et al., 2016, 2017), the ice origin and in-cloud updraught are difficult to determine from observations at a global scale and over a significant period of time. A recent classification (Gryspeerdt et al., 2017b) has shown some skill at determining these quantities when compared to a convection permitting model and is used in this work to account for this issue.

Second, the $N_i$ is a difficult property to measure at a global scale. Aircraft measurements are limited in space and time and have been afflicted by shattering of crystals on the tips of measurement probes, casting doubt on some earlier measurements of the $N_i$ (McFarquhar et al., 2007; Jensen et al., 2009; Korolev et al., 2013). Additionally, due to the highly variable nature of cirrus clouds and their strong sensitivity to environmental conditions, it can be difficult to separate the relative roles of aerosol and dynamics (Gayet et al., 2004).

The retrieval presented in part one of this work (Sourdeval et al., submitted) has demonstrated that the $N_i$ can be retrieved using a combined radar-lidar retrieval and that this compares well to new in-situ aircraft measurements where shattering is accounted for. Combined with simultaneous retrievals of the ice water content, this allows the global distribution of the $N_i$ and the factors that control it to be investigated. Using reanalysis aerosol concentrations and a proxy for the INP concentration, the

impact of aerosols on high clouds is also investigated, highlighting avenues for future research into cirrus cloud processes.

## 2 Methods

The $N_i$ dataset used in this work (DARDAR-LIM) has been described in detail in part one of this work (Sourdeval et al., submitted), so only the main features are outlined here. The DARDAR-LIM product is based on the DARDAR retrieval (Delanoë and Hogan, 2010), a combined raDAR-liDAR retrieval of ice cloud water content (IWC) and ice crystal effective radius using

data from the CloudSat and CALIPSO satellites at approximately 13:30 local solar time. Only daytime data from the period 2006-2013 is used in this work due to constraints in the reanalysis data availability. The properties are retrieved at a horizontal resolution of 1.7 km and a vertical resolution of 60 m. Both the DARDAR IWC and the DARDAR-LIM $N_i$ retrieval compare favourably to in-situ aircraft data (Deng et al., 2013; Sourdeval et al., submitted), with the best agreement at temperatures below -30°C, where the retrievals are more accurate due to the dominance of uni-modal ice crystal size distributions and reduced

ambiguity in the cloud phase.

To investigate the controls on ice crystal nucleation, a more in-depth study is performed of the $N_i$ near the cloud top ($N_{i(top)}$). As the cloud top is the location of the coldest temperature in the cloud, it has the highest theoretical nucleation rates. Although the cloud top is close to the nucleation region in wave clouds (e.g. Spichtinger and Gierens, 2009a), this is not always the case and the $N_{i(top)}$ can be rapidly reduced by differential sedimentation and entrainment (e.g. Jensen et al., 2013a). However, as the

coldest temperature, it provides a limitation on the maximum nucleated $N_i$ within the cloud, limiting the impact of temperature variability due to the vertical extent of the cloud. The cloud top is taken to be the top 120 m of the cloud and only the uppermost cloud layer (in multi-layer situations) is used to avoid issues with ice being seeded by ice crystals falling from overlying layers. The data is also restricted to locations where the retrieval has gone through at least two iterations, limiting the impact of prior assumptions about the cloud structure.

Four main controls on the $N_{i(top)}$ are considered in this work: temperature, cloud-scale updraught, ice origin and aerosol. Temperature data in this study is taken from the ECMWF ERA Interim reanalysis (Dee et al., 2011). Information about the cloud-scale updraughts and the ice-origin (liquid/ice) cannot yet be obtained directly at a global scale using satellites. To provide an indication of these cloud properties, the classification from Gryspeerdt et al. (2017b) is used. This classification is based on the assumed cirrus source (orographic, frontal, convective or synoptic) and determined at 13:30 local solar time using

cloud retrievals from the MODIS instrument (Platnick et al., 2017) and reanalysis data. This classification selects orographic clouds by assuming the product of the topographic variation and the windspeed is related to the in-cloud updraught, similar to global climate model parametrisations (e.g. Joos et al., 2008). The Oro2 and Oro1 regimes are the ones with the highest and second highest sextiles of the parametrised in-cloud updraught. Frontal and convective regimes are selected as connected

regions of high level cloud that intersect with reanalysis fronts and regions of large-scale updraught respectively. Finally, the synoptic regime is taken as a residual, with clouds being considered synoptic if they do not fit any of the other classes. Through comparisons with convection permitting model data and classifications based on reanalysis data, this classification has been shown to provide useful information on the cloud scale updraughts and the ice origin. While not a direct retrieval these properties, it does allow some inferences to be made regarding the response of the $N_i$ to these factors. The results in this paper are restricted to daytime data, which in turn restricts it to 13:30 LST due to the orbit of the satellites used to construct the DARDAR-LIM dataset.

To investigate a possible aerosol link to $N_i$, we use the "monitoring atmospheric composition and climate" (MACC) aerosol re-analysis product (Morcrette et al., 2009; Benedetti et al., 2009), which assimilates MODIS aerosol optical depth (AOD) retrievals into the ECMWF integrated forecast system. The MACC product provides altitude information for aerosols along with speciation information. Although the MACC speciation has not yet been validated, the MODIS cloud droplet number concentration shows a stronger sensitivity to hydrophilic aerosol types that the hydrophobic ones, suggesting that the MACC speciation conveys useful information about the aerosol type (Gryspeerdt et al., 2016). Further from sources, where ageing and other assumptions come into play, the speciation may be less accurate. In the upper troposphere, liquid aerosol is thought to be the dominant aerosol component (Kojima et al., 2004), although recent studies have noted that glassy organic aerosols re abundant in the upper atmosphere and can act as INP at temperatures below -55°C (Murray, 2008; Wilson et al., 2012). This work takes the MACC total mass concentration as a measure of the liquid aerosol concentration (high (low) aerosol is more (less) than 6 $\mu\mathrm{g\,m^{-3}}$), with the caveat that this measure of aerosol may shift towards a measure of INP at the coldest temperatures considered in this work.

The response of the $N_i$ to aerosol is likely to vary by aerosol type (Pruppacher and Klett, 1997). Although MACC provides a dust speciation, it is not clear whether this can be used to determine the presence of INP. Previous studies have suggested that high altitude aerosol may be responsible for glaciating clouds between 0°C and -35°C (Choi et al., 2010; Kanitz et al., 2011; Zhang et al., 2012; Tan et al., 2014). Based on this previous work, the glaciation of clouds between 0°C and -35°C is used as a proxy for the occurrence of INP.

Cloud phase is determined using the DARDAR phase detection algorithm (v1.1.4; Delanoë and Hogan, 2010). This uses the different response of lidar backscatter and radar reflectivity to liquid and ice hydrometeors to identify glaciated clouds. Clouds with a peak in lidar backscatter at the cloud top are treated as liquid or mixed phase and those with only a strong radar return are treated as glaciated. Experience suggests that the retrieved phase can be unreliable for clouds less than 600 m thick, so these are excluded from this part of the analysis.

Using this cloud top phase product, a "glaciation index" is developed using the phase retrievals between 0 and -35°C. As approximately half of cloud tops are glaciated at -20°C, glaciated clouds with a top temperature warmer than -20°C are identified as "warm ice", and liquid topped clouds colder than -20°C are correspondingly "cold liquid". The "warm ice" pixels are taken to indicate the presence of INP within 100 km (the approximate spatial scale of aerosol variability from Weigum et al., 2012), whilst the "cold liquid" ones are taken to indicate a relatively INP-free environment. If both (or neither) are detected within 100 km, that pixel is excluded from the analysis. The cloud phase is only used for the uppermost cloud layer when

determining this INP proxy to reduce the impact of overlying ice clouds seeding ice in lower layers. In addition, regions with nearby higher cloud layers (those within a 10:1 glide-slope) are also excluded from the "glaciation index". To reduce the impact of random errors in the phase retrieval, two or more neighbouring pixels are required for a detection of "warm ice" or "cold liquid". This "glaciation index" is produced for each DARDAR vertical profile and is then used as a proxy for the occurrence

5    of INP at temperatures colder then -35°C in that profile. The use of cloud glaciation as an INP proxy for colder temperatures in the atmosphere assumes that cloud glaciation is correlated to INP between 0 and -35°C and that there is sufficient vertical correlation in INP occurrence. These assumptions are discussed in section 4.2.1.

This combination of reanalysis temperature and aerosol data, along with previously determined clouds regimes and a proxy for INP are used globally for daytime data over the period 2006-2013 to investigate the role of different processes on the $N_i$.

10   ## 3   Results

### 3.1   Global $N_{i(top)}$ distribution

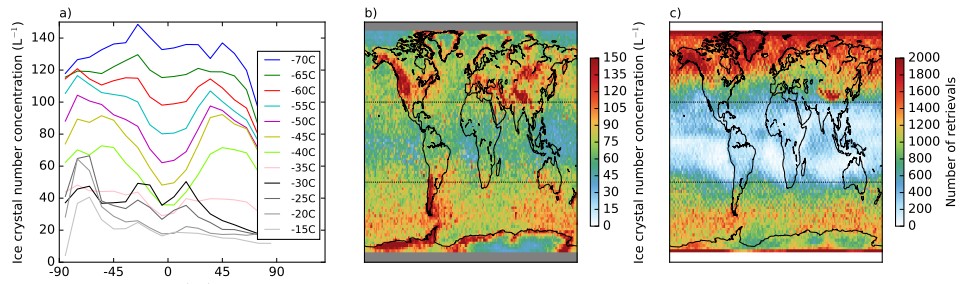

**Figure 1.** a) The zonal mean DARDAR-LIM cloud top $N_i$ ($N_{i(top)}^{5\mu m}$) for crystals larger than 5 $\mu$m as a function of temperature from DARDAR-LIM data for the period 2006-2013. Temperatures warmer than -35°C are in greyscale. b) The mean $N_{i(top)}^{5\mu m}$ at -50°C. Grey indicates missing data. c) The number of cloud top retrievals at -50°C. Zonal means and maps of $N_i$ are available in part one.

The global $N_{i(top)}$ distribution for crystals larger than $5\mu$m ($N_{i(top)}^{5\mu m}$) displays several features that highlight the role of different cloud processes in controlling the $N_{i(top)}^{5\mu m}$. The zonal mean $N_{i(top)}^{5\mu m}$ (Fig. 1a) shows a strong temperature dependence, with significant increases in the $N_{i(top)}^{5\mu m}$ as the temperature decreases from a mean of around $40\,\mathrm{L}^{-1}$ at -35°C to almost $140\,\mathrm{L}^{-1}$

15   at -70°C. This temperature dependence is particularly strong at temperatures colder than -40°C in both the northern and the southern mid-latitudes. There is also a strong $N_{i(top)}^{5\mu m}$ increase in the tropics, although the initial increase in $N_{i(top)}^{5\mu m}$ at -40°C is weaker. Although the $N_i$ produced by heterogeneous nucleation should also increase as temperatures decrease due to increasing INP concentrations (DeMott et al., 2010), this strong increase in $N_{i(top)}^{5\mu m}$ at -40°C along with a continuing $N_i$ increase at colder temperatures is indicative of homogeneous nucleation, which is only significant temperatures below around -35°C.

At temperatures warmer than -35°C, the mean $N_{i(top)}^{5\mu m}$ is relatively small, especially in the northern hemisphere where it averages less than $50\,L^{-1}$. This is expected from heterogeneous nucleation, where the $N_{i(top)}^{5\mu m}$ is limited by available INP. The mean $N_{i(top)}^{5\mu m}$ is much larger in the southern hemisphere and the tropics, although this is skewed by the long tail of the $N_{i(top)}^{5\mu m}$ distribution (Fig. 2). A phase misclassification, with liquid topped cloud being mistaken for ice cloud might explain this

hemispheric contrast, due to the large amounts of supercooled water in the southern hemisphere (Choi et al., 2010). A strong lidar backscatter at the cloud top would lead to a large retrieved $N_{i(top)}^{5\mu m}$ (if it was mis-classified as an ice cloud). As liquid topped clouds at sub-zero temperatures are more common in the southern hemisphere, this would result in an erroneously large mean $N_i$ in the southern hemisphere.

There are large geographical variations in $N_{i(top)}^{5\mu m}$. At -50°C, the $N_{i(top)}^{5\mu m}$ is strongly affected by the topography (Fig. 1b).

High $N_{i(top)}^{5\mu m}$ values are retrieved in mountainous regions over land and around the edge of the Antarctic ice sheet, similar to results from orographic cirrus parametrisations in global climate models (e.g. Joos et al., 2008; Barahona et al., 2017) and other satellite retrievals (Mitchell et al., 2016, 2018). This is consistent with a high $N_{i(top)}^{5\mu m}$ being generated through orographic uplift, which can generate the strong updraughts and high supersaturations required for homogeneous nucleation. While it is possible that the increased $N_{i(top)}^{5\mu m}$ is due to an increase in the vertical transport of INP, the lack of a similar pattern in

the cloud supercooled fraction at -20°C (Choi et al., 2010) makes this explanation unsatisfactory. The $N_{i(top)}^{5\mu m}$ in the tropics is comparatively low, even in regions of significant topography such as the Ethiopian Highlands. This is due to low wind speeds in the tropics reducing the in-cloud orographic updraught, similar to the GCM results of Joos et al. (2008). The high orographic $N_{i(top)}^{5\mu m}$ also partially explains the hemispheric asymmetry in $N_{i(top)}$ in the mid-latitude and polar regions, due to the high $N_{i(top)}^{5\mu m}$ generated by orographic clouds over the Andes and around the edge of the East Antarctic ice sheet. The $N_{i(top)}^{5\mu m}$

in the tropics is significantly lower than the $N_i^{5\mu m}$ at -50°C (Part 1). This is partly due to the low number of cloud tops at this temperature in the tropics, meaning that there is a clear sampling bias. Additionally, the cloud top temperature plays an important role in determining the $N_i^{5\mu m}$, giving it a much weaker temperature dependence. This temperature dependence is investigated further in the following two subsections.

## 3.2    Cloud regime dependence

The location map and temperature dependence of the $N_{i(top)}^{5\mu m}$ (Fig. 1) and the results from part one hint that there may be a significant regime dependence in the $N_{i(top)}$, in particular a strong role for orographic clouds and a possible role for convective clouds, given the low $N_{i(top)}$ in the tropics. Separating the $N_{i(top)}$ data by regime using the classification of Gryspeerdt et al. (2017b) allows this dependence to be independently investigated. Due to the strong temperature dependence and the large variability of the $N_{i(top)}$, joint probability histograms, showing the probability of a $N_{i(top)}$ retrieval at a given temperature are

shown in Fig. 2. Following the results of part one, the $N_{i(top)}$ is investigated for crystals bigger than $5\,\mu m$ ($N_{i(top)}^{5\mu m}$) and $100\,\mu m$ ($N_{i(top)}^{100\mu m}$). With a minimum size of $5\,\mu m$, $N_{i(top)}^{5\mu m}$ typically lines up with the smallest sizes measured by in-situ instruments, while with a larger minimum size, $N_{i(top)}^{100\mu m}$ covers a size range where less shattering is expected and where the normalised size distribution performs well (Delanoë et al., 2005; Sourdeval et al., submitted). As noted in the previous section, the skewed

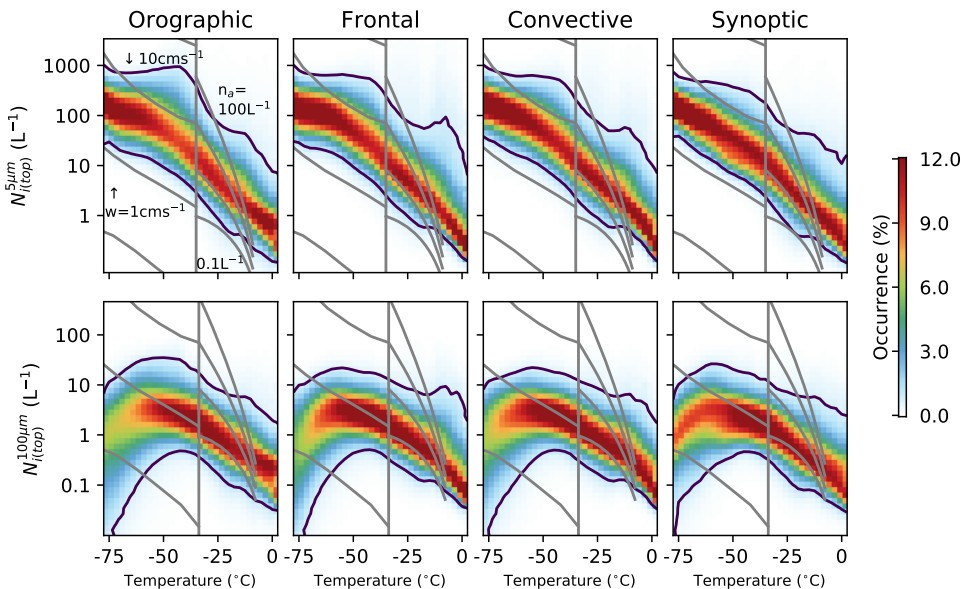

**Figure 2.** The conditional $N_{i(top)}$ ($L^{-1}$) for 5°C temperature bins for each of the main cloud regimes (Orographic, Frontal, Convective, Synoptic) from Gryspeerdt et al. (2017b, O2, F, C, S). The top row shows the $N_{i(top)}$ for particles greater than 5$\mu$m ($N_{i(top)}^{5\mu m}$) and the second greater than 100$\mu$m ($N_{i(top)}^{100\mu m}$). The columns are normalised so that they sum to 100%. The vertical line is drawn at -35°C - approximately the homogeneous nucleation threshold temperature. At temperatures warmer than -35°C, the gridlines show the INP numbers predicted by the DeMott et al. (2010) parametrisation for 0.1, 1, 10, 100$L^{-1}$ aerosols >0.5$\mu$m. The gridlines at temperatures below -35°C are the $N_i$ values following Koop et al. (2000) for 1, 10 and 100 cm s$^{-1}$ updraught speeds for a mean pressure and an aerosol particle number of 300 cm$^{-3}$, following (Krämer et al., 2009). The regime names and definitions are given in Table 1 of Gryspeerdt et al. (2017a). Note the log scale for $N_{i(top)}$.

$N_{i(top)}$ distribution makes a simple linear average complicated to interpret. For the remainder of this work, normalised joint histograms are used, showing the probability of finding a particular $N_i$, given that a certain temperature has been observed.

There are a number of broad similarities between the regimes. Each regime shows a very similar increase in $N_{i(top)}^{5\mu m}$ with decreasing temperature, with the decrease becoming weaker at very colder temperatures, rising to around $100\,L^{-1}$ at -75°C. While this is larger than the $N_i$ values reported from many measurements of tropical tropopause layer cirrus (e.g. Jensen et al., 2013a), this may be due to sampling differences between the satellite and in-situ measurements, with some of the thinnest clouds being missed by the DARDAR retrieval. It is also possible that uncertainties in the shape of the particle size distribution (PSD) can lead to overestimations of $N_{i(top)}^{5\mu m}$ by as much as a factor of two (Sourdeval et al., submitted). The temperature dependence is similar to that observed during the SPARTICUS and MACPEX campaigns (Jensen et al., 2013b; Muhlbauer et al., 2014), although the temperature dependence is stronger than that observed in Krämer et al. (2009) where the $N_i$ was sample in cloud, rather than at the cloud top. However, if the satellite data is sampled in a manner similar to previous work, it

reproduces the in-situ results (Sourdeval et al., submitted), giving confidence in the magnitude and temperature dependence of the results presented here. There is evidence of possible retrieval errors, as both the orographic and convective regimes have a small number of retrievals of over $30\,\mathrm{L}^{-1}$, with some as high as $50\,\mathrm{L}^{-1}$, around -15°C. This suggests that the possible phase misclassification and the high $N_{i(top)}$ values observed in the zonal mean are more common in certain regimes.

All of the regimes also show a peak in the highest $N_{i(top)}^{5\mu m}$ percentiles at temperatures just colder than -35°C. The strength varies by regime, with the orographic regime showing a stronger peak and only a weak peak being observed in the frontal and convective regimes. The peak is barely present in the synoptic regime, where the $N_{i(top)}^{5\mu m}$ is located on the trend in $N_{i(top)}^{5\mu m}$ present at other temperatures. An increase in the largest $N_{i(top)}^{5\mu m}$ values at this temperature is consistent with homogeneous nucleation, either through an increase in the freezing of liquid droplets or by increased homogeneous nucleation through the

freezing of unactivated aqueous haze particles. At these colder temperatures, the $N_{i(top)}$ is roughly parallel with the contours of $N_i$ expected through homogeneous nucleation at a constant updraught (Krämer et al., 2009), especially in cases where the peak in $N_{i(top)}^{5\mu m}$ at -40°C is small (Fig.2). Warmer than -35°C, the $N_{i(top)}^{5\mu m}$ is broadly consistent with the number of INP predicted by the DeMott et al. (2010) parametrisation, but the $N_{i(top)}^{5\mu m}$ becomes increasingly large compared to the number of INP as the temperature increases. As the DARDAR-LIM retrieval has not been evaluated at this temperature, it is unclear if this is a real

effect, or if it is due to the possible phase classification issue mentioned previously.

The variation in the strength of this peak is clearly seen when comparing the Oro 2 and Oro 1 regimes (the highest and second highest sextiles of the estimated in-cloud updraught) in Fig. 3a. While there is little difference between the regimes at warmer temperatures, below -35 °C there is a strong increase in the $N_{i(top)}^{5\mu m}$ in the Oro 2 regime. This increase peaks at about -50 °C, reducing and almost disappearing at the coldest temperatures studied. The high $N_{i(top)}^{5\mu m}$ retrieved in these clouds and

the strong dependence on the in-cloud updraught explain the geographical pattern shown in Fig. 1b, where high $N_{i(top)}^{5\mu m}$ are observed in mountainous regions. A high $N_{i(top)}^{5\mu m}$ in these regimes is supported by results from previous in-situ studies, where large $N_i$ values were recorded in orographic clouds (Jensen et al., 1998; Field et al., 2001; Baker and Lawson, 2006).

It is possible that this increased $N_{i(top)}^{5\mu m}$ is the result of increased aerosol concentrations carried to lower temperatures in the stronger updraughts of the Oro 2 regime. However, the lack of a difference in $N_{i(top)}^{5\mu m}$ between the regimes at temperatures

warmer than -35 °C indicates that an increase in INP is not driving this change in $N_{i(top)}^{5\mu m}$, which in turn makes it less likely that this change is due to a change in liquid aerosol. As the Oro regimes are defined by the estimated in-cloud updraught (Gryspeerdt et al., 2017b), the difference between the regimes shown in Fig. 3 is likely due to a change in the updraught environment impacting freezing processes.

A change in the updraught environment could modify the $N_{i(top)}^{5\mu m}$ by changing the likelihood of homogeneous nucleation,

through an either through allowing more liquid droplets to reach temperatures where they can freeze homogeneously or by increasing the nucleation of haze droplets (Koop et al., 2000). These processes cannot be easily distinguished in the current study, although the lack of a significant occurrence of liquid-topped cloud in orographic regions (Tan et al., 2014) suggests that an increased cloud droplet number concentration is not the leading contributor to the increase in $N_{i(top)}^{5\mu m}$. This strong response to updraught changes would support previous studies that highlighted the updraught limited nature of many cirrus clouds (Kay

and Wood, 2008; Barahona and Nenes, 2008). A larger difference exists between the frontal and synoptic regimes (Fig. 3b),

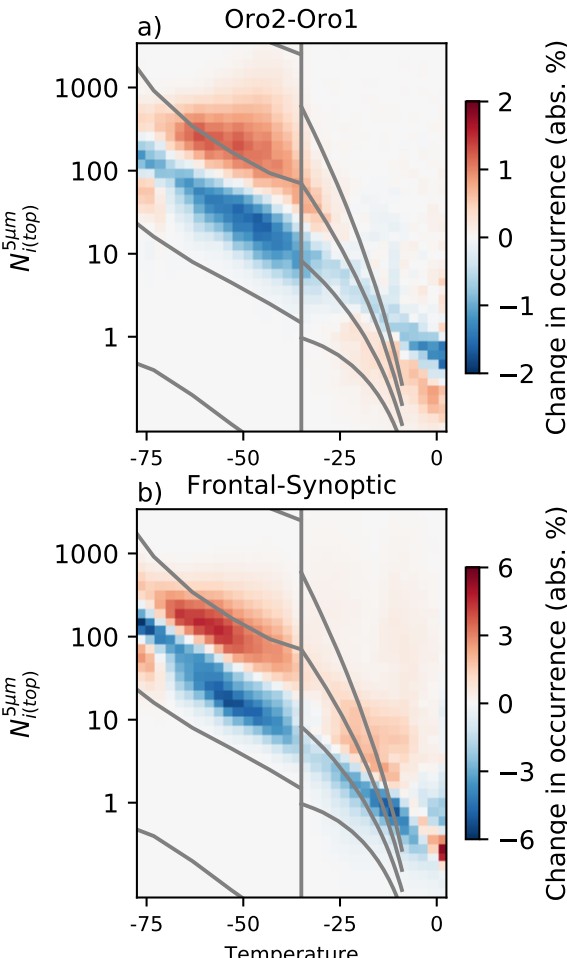

**Figure 3.** a)The difference the the $N_{i(top)}^{5\mu m}$ as a function of temperature between the Oro 2 and Oro 1 regimes (highest and second highest sextiles of estimated in-cloud updraught). Red above blue at a given temperature indicates an increased $N_{i(top)}^{5\mu m}$ in the Oro 2 regimes compared to Oro 1. b) The difference between the frontal and synoptic regimes. Note the different colourscale from (a).

indicating that the magnitude of this updraught effect could be stronger than is shown here. However, the difference between the frontal and synoptic regimes cannot be easily attributed to updraught variations.

The temperature dependence of crystals larger than $100\,\mu$m ($N_{i(top)}^{100\mu m}$) at the cloud top displays a different pattern (Fig. 2, bottom row). While $N_{i(top)}^{100\mu m}$ and temperature are negatively correlated at warmer temperatures, the $N_{i(top)}^{100\mu m}$ reaches a peak at
5    around -50 °C and there is a decrease in the $N_{i(top)}^{100\mu m}$ as temperatures reduce further, with the strongest decrease observed in the orographic regime. This is consistent with a shift towards smaller ice crystals at the cloud top with colder temperatures. The synoptic regime shows a weaker decrease in $N_{i(top)}^{100\mu m}$, indicating a slightly larger role for larger ice crystals in this regime. This

shift towards smaller crystals at the cloud top is expected due to slower depositional growth and aggregation of ice crystals at colder temperatures resulting in crystals precipitating from the cloud top region before they grow larger than $100\,\mu$m.

### 3.3  The $N_i$ within clouds

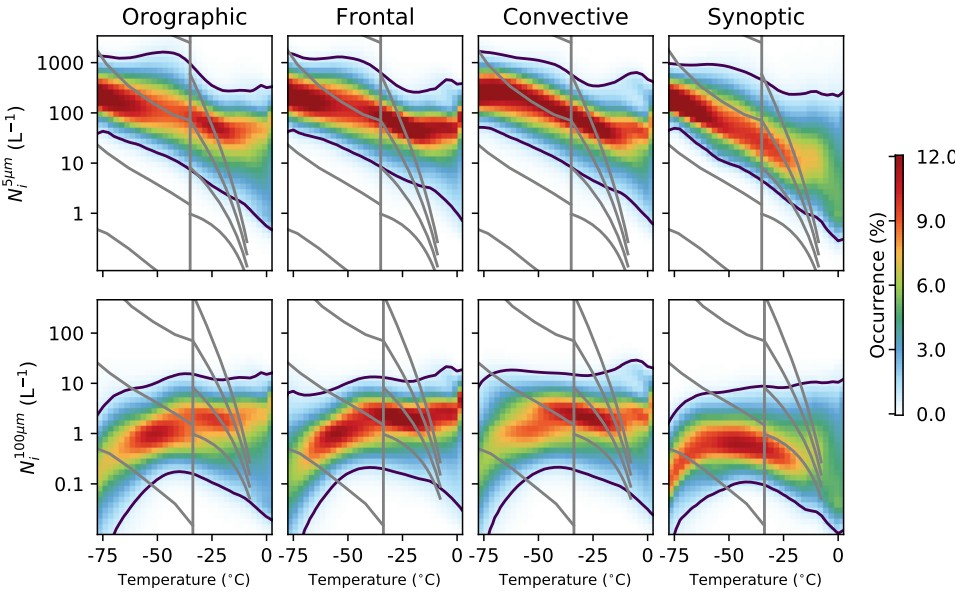

**Figure 4.** As Fig. 2 but using the $N_i$ from throughout the cloud, rather than just the cloud top. The temperature scale is the temperature of the $N_i$ retrieval, rather than that of the cloud top.

The behaviour of the $N_i$ within clouds as a function of temperature displays some significant contrasts to the $N_{i(top)}^{5\mu m}$ (Fig. 4).
While all of the regimes show an increase in the $N_i^{5\mu m}$ with decreasing temperature, this increase is much weaker than the increase in the $N_{i(top)}^{5\mu m}$. Similarly, although the peak that is visible in the $N_{i(top)}^{5\mu m}$ at about -40 °C is still visible in $N_i^{5\mu m}$ in the orographic and frontal regimes, it is much weaker than the peak observed at the cloud top. The synoptic regime has the strongest temperature dependence of all of the regimes. One explanation is the lower average cloud depth, such that the $N_i^{5\mu m}$ retrieved is often closer to the cloud top than in the other regimes. In all of the regimes, the $N_i^{5\mu m}$ is much larger in-cloud than at the cloud top for temperatures warmer than -30 °C, with values up to $100\,\mathrm{L}^{-1}$ being commonly observed. The smaller $N_i^{5\mu m}$ values that are more typically observed in the synoptic regime than the other regimes suggest that seasonal variations of the cloud classes (Gryspeerdt et al., 2017b) are likely responsible for the seasonal variations in $N_i^{5\mu m}$ observed in part one.

Similar to the $N_i^{5\mu m}$, the temperature dependence of the $N_i^{100\mu m}$ is very different internally within clouds compared to at cloud tops. The temperature dependence is much weaker, with almost no temperature dependence at temperatures warmer than -50 °C. There is a decrease in the $N_i^{100\mu m}$ at the lowest temperatures, similar to the decrease in the $N_{i(top)}^{100\mu m}$ seen in Fig. 2 and

is explained by the retrievals at these temperatures being closer to the cloud top than at warmer temperatures. The synoptic regime has the lowest $N_i^{100\mu m}$ at warmer temperatures, which may again be due to the lower geometrical thickness of clouds in this regime, such that the $N_i^{100\mu m}$ is typically located closer to the cloud top, resulting in a lower $N_{i(top)}^{100\mu m}$ for any given temperature inside a cloud.

The larger $N_i^{100\mu m}$ values at warmer temperatures mean that larger crystals comprise a larger proportion of the $N_i^{5\mu m}$, with a reduced contribution of small crystals to the $N_i^{5\mu m}$. A weaker temperature dependence of the $N_i^{5\mu m}$, especially at temperatures colder than -35 °C, is in better agreement with the results from Krämer et al. (2009), although a temperature dependence remains. It is possible that the weak temperature dependence in previous results could be due to a lack of measurements near the cloud top, where the temperature dependence is strongest. This may also explain the apparent mismatch between the INP

and $N_i$ concentration in aircraft data (e.g. Kanji et al., 2017), as the retrieved $N_{i(top)}$ values are a much closer match to the INP concentrations predicted by the DeMott et al. (2010) parametrisation than the internal $N_i$ at temperatures warmer than -35 °C (Fig. 2). Further sampling differences between the satellite and in-situ studies due to detection limits of satellite instruments and the structuring of flight campaigns may explain the remaining differences between $N_i$ determined using different methods.

### 3.4   Vertical structure of $N_i$

The $N_i^{5\mu m}$, $N_i^{100\mu m}$ and ice water content (IWC) all change significantly as a function of depth through the cloud (Fig. 5). For clouds with a top temperature between -40 and -50°C (Fig. 2), the $N_i^{5\mu m}$ continues to increase until about 500 m from the cloud top, at which point it starts to decrease again (Fig. 5, top row). The $N_i^{5\mu m}$ distribution width stays approximately constant from about 1 km into the cloud until around 2-3 km from the cloud top, when it reaches a temperature of around -30°C when liquid water can form more easily. At this point the $N_i^{5\mu m}$ distribution broadens significantly. Similar to the $N_i^{5\mu m}$, the $N_i^{100\mu m}$ also

grows quickly when moving down through the cloud, moving to a slower growth regime after the first 500 m from the cloud top. This shift in the $N_i^{100\mu m}$ growth regime is roughly coincident with the location of the $N_i^{5\mu m}$ peak. All of the regimes also show an increase in the IWC (Fig. 5, bottom row) with increasing depth in the cloud, with a sharp increase over 2.5 km from the cloud top. This sharp increase is consistent with a possible increase in ice through liquid water processes in warmer parts of the cloud.

There are some differences between the regimes. The synoptic regime has a much weaker peak in $N_i^{5\mu m}$ below the cloud top and a consequently lower $N_i^{5\mu m}$ throughout the depth of the regime. Despite having similar values at the cloud top to the other regimes, the $N_i^{100\mu m}$ and the IWC in the synoptic regime remain lower than the other regimes through the cloud, possibly due to lower in-cloud updraughts. At about 2.5 km from the cloud top, both $N_i^{100\mu m}$ and IWC increase until they are roughly comparable to the other regimes, suggesting that the signal from liquid water swamps any signal based on ice nucleation.

The peak is also temperature dependent, almost disappearing in clouds with colder tops (see supplementary information) and varying in size and location between the regimes. When the peak occurs in the synoptic regime, it is within 300 m of the cloud top in 67% of cases, compared to only 48% of cases for the frontal regime. These distances are comparable to the thickness of nucleation regions noted in Jensen et al. (2016) of between 20 and 500 m. The enhancement of the $N_i^{5\mu m}$ within this peak in the synoptic regime is also smaller, with an average peak of $130\,L^{-1}$, compared to $270\,L^{-1}$ in the frontal regime

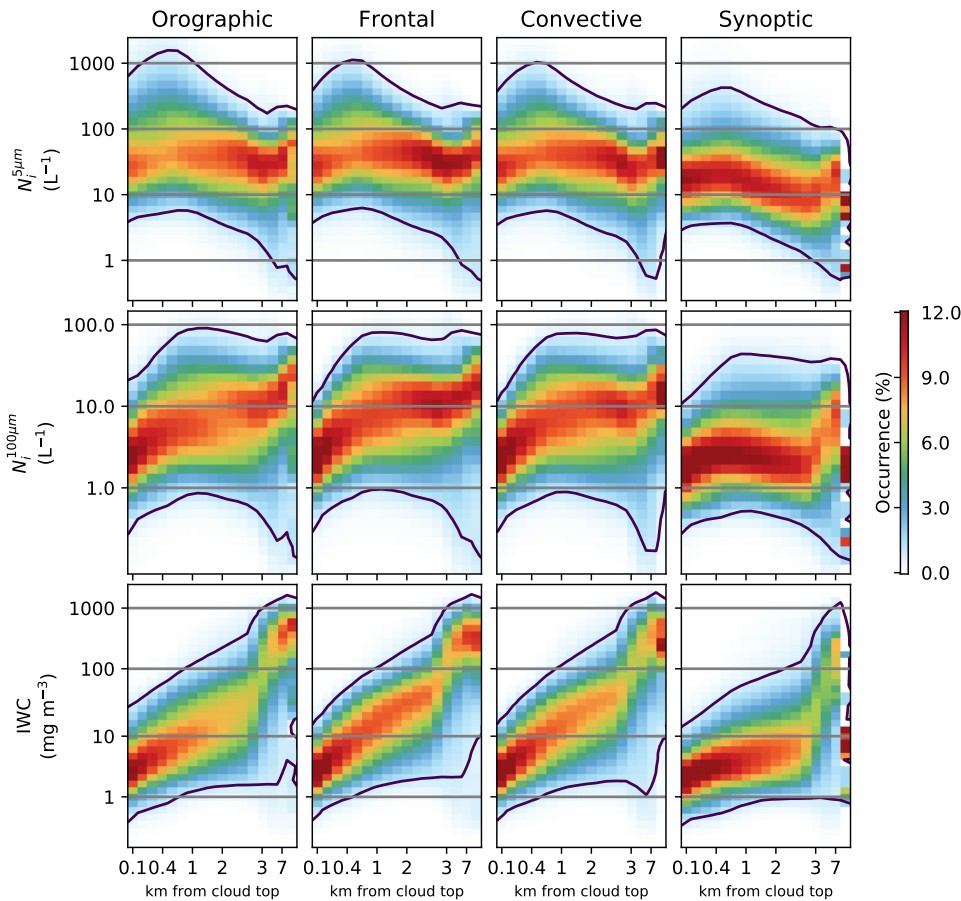

**Figure 5.** Retrieved properties as a function of the distance from the cloud top. This is for clouds with tops between -40 and -50 °C. Note the non-linear scale on the horizontal axis.

and $325\,\mathrm{L}^{-1}$ for the orographic regime. The increased strength of this peak in regimes expected to have a stronger updraught along with its location close to the cloud top may indicate a role of homogeneous nucleation. Model studies of cirrus clouds suggest that homogeneous nucleation can produce peaks in $N_i$ cloud to the cloud top (Spichtinger and Gierens, 2009a, b), with an increased $N_i$ at higher updraught velocities. The disappearance of the peak at colder temperatures gives it a similar temperature dependence to the peak in the $N_{i(top)}^{5\mu m}$ (Fig. 2) providing further supporting evidence of the impact of homogeneous nucleation on $N_i$ in this temperature range.

It is possible that the varying sensitivities of the CloudSat radar and the CALIOP lidar to crystal size and the attenuation of the CALIOP lidar in the upper levels of the cloud could be generating this vertical structure. The lower vertical resolution and sensitivity to small crystals of the radar could result in it missing the cloud top, generating a peak in the $N_i^{5\mu m}$ at the level where the retrieval starts to include radar information. However, the results in Part 1 show no evidence of a bias in the $N_i$ retrieval as a

function of the instruments contributing to the retrieval (Sourdeval et al., submitted). This is primarily due to the sensitivity of the instruments to different ice crystal size distributions. Although the lidar-only retrievals have a higher expected error, they usually only occur in cases where there is a monomodal size distribution dominated by small crystals that can be accurately constrained by the lidar alone. Additionally, the disappearance of the peak at colder temperatures indicates that it is a physical property of the clouds, rather than a property of the retrieval, as the instrument sensitivities would not be expected to strongly vary with temperature.

## 4  The relationship to aerosol

### 4.1  The relationship to liquid aerosol

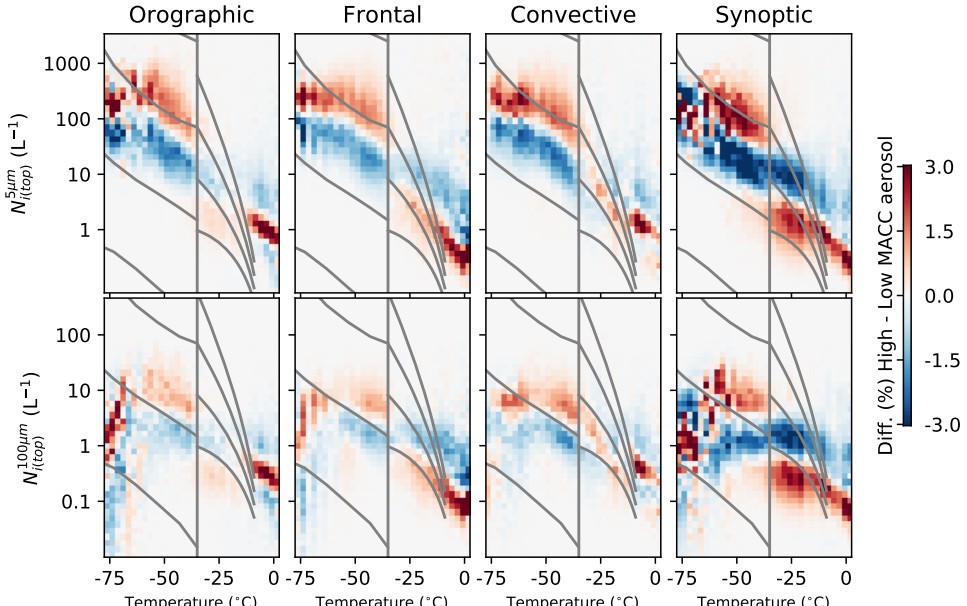

**Figure 6.** The difference in the conditional histograms between cases with high MACC total aerosol mass concentration ($>6\,\mu g\,m^{-3}$) and low total mass concentration ($<6\,\mu g\,m^{-3}$) for the four main regimes. The gridlines are the same as Fig. 2. The upper set of plots show the difference in $N_{i(top)}^{5\mu m}$ and the lower in $N_{i(top)}^{100\mu m}$. The changes sum to zero vertically, red over blue indicates an increase in the $N_{i(top)}^{5\mu m}/N_{i(top)}^{100\mu m}$ for a given temperature and regime.

Fig. 6 shows how the $N_{i(top)}^{5\mu m}$ distribution changes as a function of temperature and MACC reanalysis aerosol (used to indicate high concentrations of liquid aerosol). In most of the regimes, there is a positive relationship between MACC aerosol and $N_{i(top)}^{5\mu m}$ at temperatures below -35 °C (shown by red above blue in Fig. 6). In the synoptic regime, this positive aerosol-$N_{i(top)}^{5\mu m}$ relationship only exists for temperatures warmer than -60 °C – at temperatures colder than this, the relationship becomes weak

and noisy. In the other regimes, the positive relationship is maintained to very cold temperatures. At temperatures warmer than -35 °C, the relationship becomes a lot weaker, with almost no aerosol-$N_{i(top)}^{5\mu m}$ relationship existing in the orographic and convective regimes. In the frontal regime, there is a slight negative relationship, with a stronger negative relationship in the synoptic regime. It is possible that this negative relationship is related to a misclassification of ice and liquid at these warmer temperatures being a function of the MACC aerosol, particularly in regions where INP rich aerosol constitute a majority of the aerosol population.

The aerosol-$N_{i(top)}^{100\mu m}$ relationship shows a weaker pattern than the aerosol-$N_{i(top)}^{5\mu m}$ relationship, with the smaller enhancement of the $N_{i(top)}^{100\mu m}$ at colder temperatures in most regimes indicating a shift to smaller crystal sizes. The change in the synoptic regimes is the strongest, likely related to the strong relationship for the $N_{i(top)}^{5\mu m}$. A negative relationship between aerosol environment and crystal size has been noted in previous work (Jiang et al., 2011; Zhao et al., 2018) and often corresponds to an increase in $N_{i(top)}$, although positive relationships have been observed over the Indian Ocean (Chylek et al., 2006).

It is difficult to demonstrate causality with observed aerosol-cloud relationships, to the extent that it is not clear that this relationship is a change in $N_{i(top)}$ due to a change in aerosol. However, this strong relationship between MACC aerosol and $N_{i(top)}$ is consistent with an increased ice crystal nucleation through homogeneous nucleation, which can be sensitive to the concentration of liquid aerosol (e.g. Kärcher, 2002). In situations where the $N_{i(top)}$ is primarily determined by the freezing of liquid droplets, an increase in cloud droplets in high aerosol regions could also lead to an increased $N_{i(top)}$, although the number of droplets frozen is relatively insensitive to the total number of liquid droplets (Kärcher and Seifert, 2016). As with the impact of in-cloud updraught on $N_{i(top)}$, further investigation is required to determine if one of these mechanisms is dominant. As liquid water has been found in clouds at temperatures as cold as -40°C, increased droplet freezing cannot be ruled out, even though many clouds are frozen before reaching this temperature (Choi et al., 2010). At colder temperatures, it seems likely that homogeneous nucleation plays a role, as liquid droplets cannot form at these temperatures. In this case, the stronger updraughts in the frontal and convective regimes are important for generating the high supersaturations in which homogeneous nucleation can occur. Changing aerosol types may also play a role at temperatures colder than -60°C, where the increasing impact of glassy aerosols may lead the aerosol to nucleate ice heterogeneously. A combination of the weak expected updraughts and the increasing ability of glassy aerosol to act as an INP at low temperatures may explain the weak aerosol-$N_{i(top)}$ relationship in the synoptic regime below -60°C. While there is a clear relationship in Fig. 6, the change in the mean $N_{i(top)}$ is small, even for this large aerosol perturbation. At -50°C, the mean $N_{i(top)}^{5\mu m}$ increases from around 140 to 175 $L^{-1}$, an increase of 25%. Much of this change is driven by changes in the high updraught tail of the distribution, producing a 25% change in $N_{i(top)}$ at -50°C would require an updraught in excess of $1 ms^{-1}$ (Kärcher, 2002). While plausible for the convective and orographic regimes (Gryspeerdt et al., 2017a), the large updraughts required to generate such a sensitivity may indicate that this relationship is affected by an updraught mediated covariation.

## 4.2 The relationship to an INP proxy

The sparse nature of INP measurements (e.g. Mamouri and Ansmann, 2016) and the high sensitivity of the $N_i$ to low INP concentrations means that it is difficult to use retrieved aerosol properties to investigate the effect of INP on the $N_{i(top)}$. To

avoid this issue, the glaciated fraction of clouds lower in the atmosphere (-20°C) is used as a proxy for the presence of INP at other levels in the atmosphere.

### 4.2.1 Cloud glaciation and INP

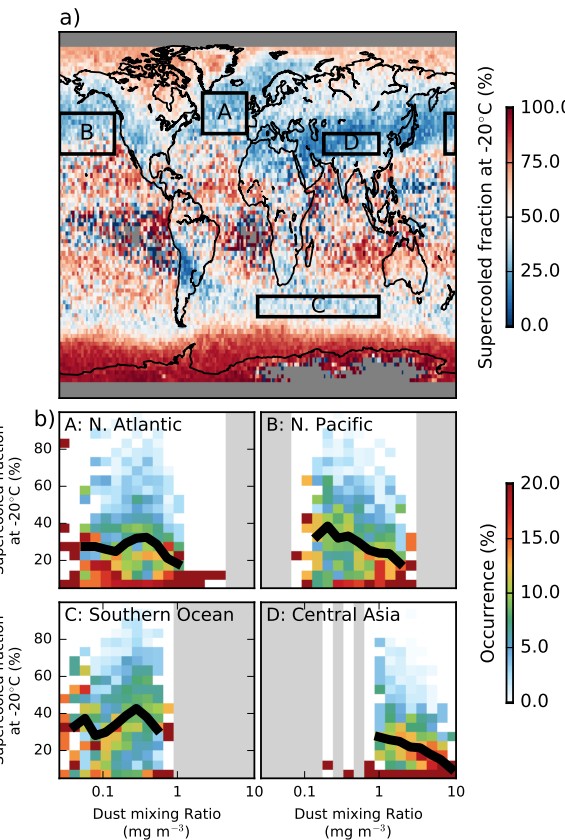

**Figure 7.** a) The DARDAR supercooled fraction at -20°C, defined as the fraction of DARDAR cloud top phase retrievals between -17.5 and -22.5 °C from 2006 and 2013 that are classed as liquid. b) The conditional probability of observing a daily mean supercooled fraction, given a specified MACC dust mixing ratio for the regions specified in (a). The black line shows the mean supercooled fraction for each aerosol bin.

The addition of the CloudSat data in the DARDAR product allows smaller quantities of ice to be detected than the lidar-only studies, but produces a very similar pattern of cloud glaciation (Fig. 7) to the previous CALIOP studies (Choi et al., 2010; Tan et al., 2014). Calculating the supercooled fraction as the number of liquid retrievals divided by the total number of liquid and ice retrievals between -17.5 °C and -22.5 °C from 2006 and 2013, using only the top layer of clouds where the layer is more than 600 m thick.

There is a strong hemispheric contrast with a higher glaciated fraction over the northern hemisphere and a high supercooled fraction over the southern ocean and Antarctica, as observed in previous aircraft (Huang et al., 2012) and satellite studies (Choi et al., 2010; Tan et al., 2014). High glaciated fractions are observed over desert locations in central Asia and Iran, stretching across the North Pacific to the Americas. This is consistent with previous studies suggesting that dust is a good INP. Previous

studies have found Asian dust over California, suggesting that transport across the Pacific is not unexpected (Creamean et al., 2013). There is also a significant proportion of glaciated cloud downwind of the Andes, which appears to originate near the Altiplano and Patagonia. These are sources of high altitude dust (Ginoux et al., 2012) and would support the hypothesis that high altitude dust is able to glaciate clouds. While glaciated cloud in this region has been previously noted (Choi et al., 2010), the lower resolution of the previous study made it difficult to determine the source of possible INP. The longer dataset

and increased spatial resolution of Fig. 7a make the source in the upper Andes much clearer. Although southern Africa and Australia are also sources of dust (Ginoux et al., 2012), this dust is emitted at lower altitudes, which would explain the lower glaciated fractions downwind of these regions.

The origin of the glaciated region over the north Atlantic is less clear, as there are not many local sources of high level dust in the region. It is possible that the dust here has been transported across the Sahara and lofted by cyclone systems crossing

the Atlantic. It is also possible the black carbon or ash (Grawe et al., 2016) from North America may act as an INP. This might explain the lower supercooled fraction over Siberia, where black carbon from fires typically occurs without the other aerosols that are found in industrial pollution, allowing it to act as an INP (Rosenfeld et al., 2011) despite the low amounts of high level dust in this region.

The ice nucleating impact of dust for driving the cloud glaciated fraction is supported by comparing the cloud glaciated

fraction to reanalysis aerosol fields(Fig. 7b). Strong negative correlations between the occurrence of supercooled liquid cloud at -20°C and the mass concentration of reanalysis dust (Fig. 7b) are observed in some regions, with glaciated cloud dominating at high mass concentrations of reanalysis dust. However, this correlation varies by region. A stronger relationship is found in regions close to dust sources, such as over the N. Pacific (B) and central Asia (D). The relationship is much weaker in the N. Atlantic (A) and the southern ocean (C) where the dust is further from source.

The stronger dust-glaciation relationship close to the dust source, where the MACC aerosol speciation is best suggests that the supercooled fraction of clouds at -20°C is strongly related to the occurrence of INP. The weaker relationship further from source suggests that although the MACC speciation has been shown to provide useful information on aerosol type, this speciation is less reliable further from source. This is supported by results in liquid clouds, where the dust optical depth-cloud droplet number concentration relationship becomes stronger further from dust sources (Gryspeerdt et al., 2016).

Due to the reduced speciation skill from MACC far from dust sources, the occurrence of glaciated cloud at -20°C is used as a proxy for the occurrence of INP instead of the reanalysis aerosol. This relies on two assumptions:

1. Cloud glaciation at -20°C is related to INP at -20°C

2. INP at -20°C is correlated to INP at other temperatures

Based on the relationship to MACC dust aerosol, the first assumption holds in many cases. Although the second assumption is tenuous, previous studies have found similar relationships between cirrus cloud properties and both column and layer AOD (Zhao et al., 2018), similar to model results showing a significant correlation between high altitude CCN concentration and column AOD (Stier, 2016). Significant vertical aerosol autocorrelation has also been observed in global climate models (Weigum, 5  2014). Additionally, there is very unlikely to be a negative correlation between the INP at the two temperature levels, with the worst case being no correlation. As such, the relationship between the $N_i$ and the INP proxy is unable to give a quantitative result for the impact of INP on the $N_{i(top)}^{5\mu m}$, but it is able to provide a qualitative indication of the sign of the INP impact.

### 4.2.2  The INP relationship to $N_{i(top)}$

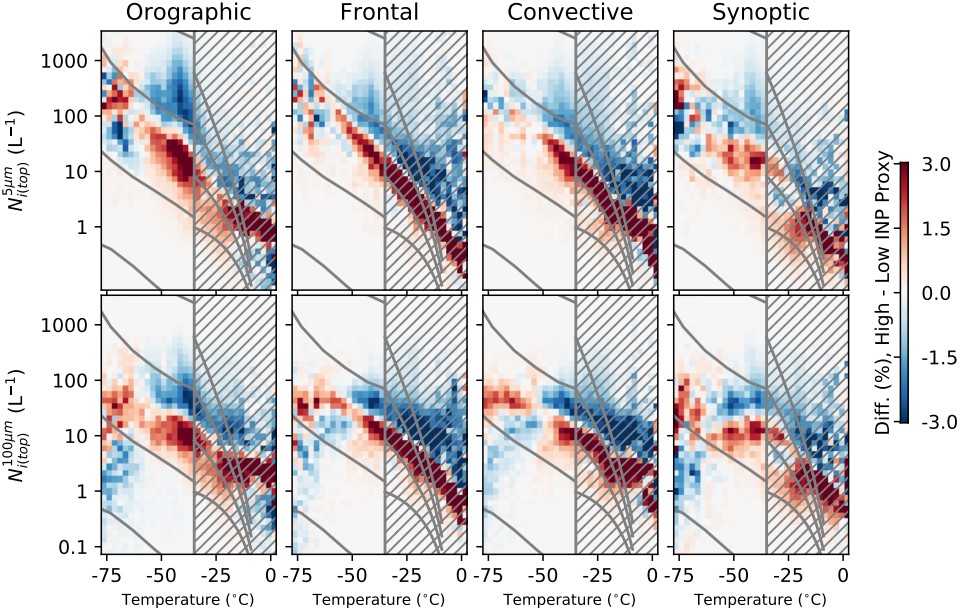

**Figure 8.** As Fig. 6 but showing the difference in the conditional histograms as a function of the INP proxy. Red indicates an increase in occurrence of a particular bin at higher inferred INP and blue a decrease, such that red above blue indicates an increase in $N_{i(top)}$ with increased INP for a given temperature. The shaded regions are likely affected by a phase misclassification at warmer temperatures.

The relationship of the $N_{i(top)}^{5\mu m}$ to the proxy for INP is shown in Fig. 8. There are a number of features that are similar 10  between the regimes, in particular the strong negative relationship between INP occurrence and $N_{i(top)}^{5\mu m}$ at temperatures warmer than -35 °C. As with the large mean $N_{i(top)}^{5\mu m}$ values shown in Fig. 1a, this may be due to liquid clouds being misclassified as ice, resulting in large $N_{i(top)}^{5\mu m}$ values being retrieved. The requirement for "warm-ice" means that supercooled liquid occurs less frequently in the high INP cases, and as such it is less likely to be misclassified as ice. The lower frequency of this misclassification then reduces the $N_{i(top)}^{5\mu m}$ and $N_{i(top)}^{100\mu m}$ in cases of high INP. The weaker $N_{i(top)}^{5\mu m}$ response in the synoptic and

orographic regimes supports this, as the misclassification in these regimes is weaker (Fig. 2). The warmer temperatures are shaded out in Fig. 8 due to the impact of this potential misclassification.

At colder temperatures, the INP-$N_{i(top)}^{5\mu m}$ relationship starts to vary between the regimes. All of the regimes show a decrease in the $N_{i(top)}^{5\mu m}$ between around -35 °C and -50 °C, the temperatures where the peak in $N_{i(top)}^{5\mu m}$ is observed connected with in-cloud updraught (Fig. 3). This decrease is strongest in the orographic regime and weakest in the synoptic regime, similar to the $N_{i(top)}^{5\mu m}$ peak observed observed in the different regimes (Fig. 2). At temperatures colder than -50 °C, the relationship becomes different again. In all of the regimes, there is an increase in $N_{i(top)}^{100\mu m}$ with increasing INP. At temperatures colder than -50 °C, the relationship becomes different again. In all of the regimes, there is an increase in $N_{i(top)}^{100\mu m}$ with increasing INP. This is consistent with an increasing number of INP shifting the size distribution towards a smaller number of larger ice crystals. In the orographic and synoptic regimes, this increase also appears in the $N_{i(top)}^{5\mu m}$, generating a positive relationship between the INP proxy and the occurrence of small ice crystals.

As with the previous section, the impact of meteorological covariations cannot be ruled out when interpreting these plots. However, they are consistent with a reduction in $N_{i(top)}^{5\mu m}$ is due to a suppression of homogeneous nucleation by INP at around -50°C. This relationship has previously been found in satellite relationships between aerosol environment and ice crystal size, with an increase in the crystal radius in situations where heterogeneous nucleation controls the $N_i$ (Chylek et al., 2006; Zhao et al., 2018). This would fit with the results in previous sections, suggesting that the $N_{i(top)}$ at this temperature range just colder than -35°C is influenced by homogeneous nucleation. This effect would only be expected in a narrow range of updraughts (Kärcher, 2002), so further work is necessary to understand the cause of this relationship.

The increase in large crystals at the coldest temperatures (below -60°C) is consistent with an INP effect on $N_{i(top)}$ if heterogeneous nucleation was dominant at these temperatures. This would fit with the results from Fig. 6, where at the coldest temperatures, there was a relatively small response of the $N_{i(top)}^{5\mu m}$ to MACC total (liquid) aerosol, suggesting that homogeneous nucleation was not controlling the $N_{i(top)}^{5\mu m}$ in synoptic cirrus. At these coldest temperatures, dust can act as an INP at very low supersaturations (as low as 105%; Möhler et al., 2006) and organic aerosol can occur in a glassy state allowing it to act as an INP. This may explain relationships consistent with heterogeneous nucleation and a classical Twomey effect at these temperatures. It is important to note that this proxy for INP relies upon the correlation between cloud glaciation at -20°C and INP at -50°C, but the absence of this correlation would produce no relationship in Fig. 8, giving some confidence to the qualitative nature of these results.

If the peak in $N_{i(top)}^{5\mu m}$ at temperatures colder than -35°C is primarily due to droplet freezing, an increase glaciated fraction at warmer temperatures could also result in this reduction of $N_{i(top)}^{5\mu m}$ with increasing INP. As the number of INP and $N_i$ warmer than -35°C is much lower than the cloud droplet number concentration, the increase in cloud glaciation could result in a reduction in the number of cloud droplets available to form ice crystals a -35°C. This would result in a negative relationship between cloud glaciation at -20°C and the $N_{i(top)}^{5\mu m}$ at colder temperatures as observed in Fig.8. As with the relationship of $N_{i(top)}$ to updraught (Fig.3) and aerosol (Fig.6), the difference between an aerosol impact on homogeneous nucleation, a change in droplet freezing or an updraught-mediated covariation (no causal effect of aerosol) cannot be distinguished by this analysis.

## 5 Vertical information propagation

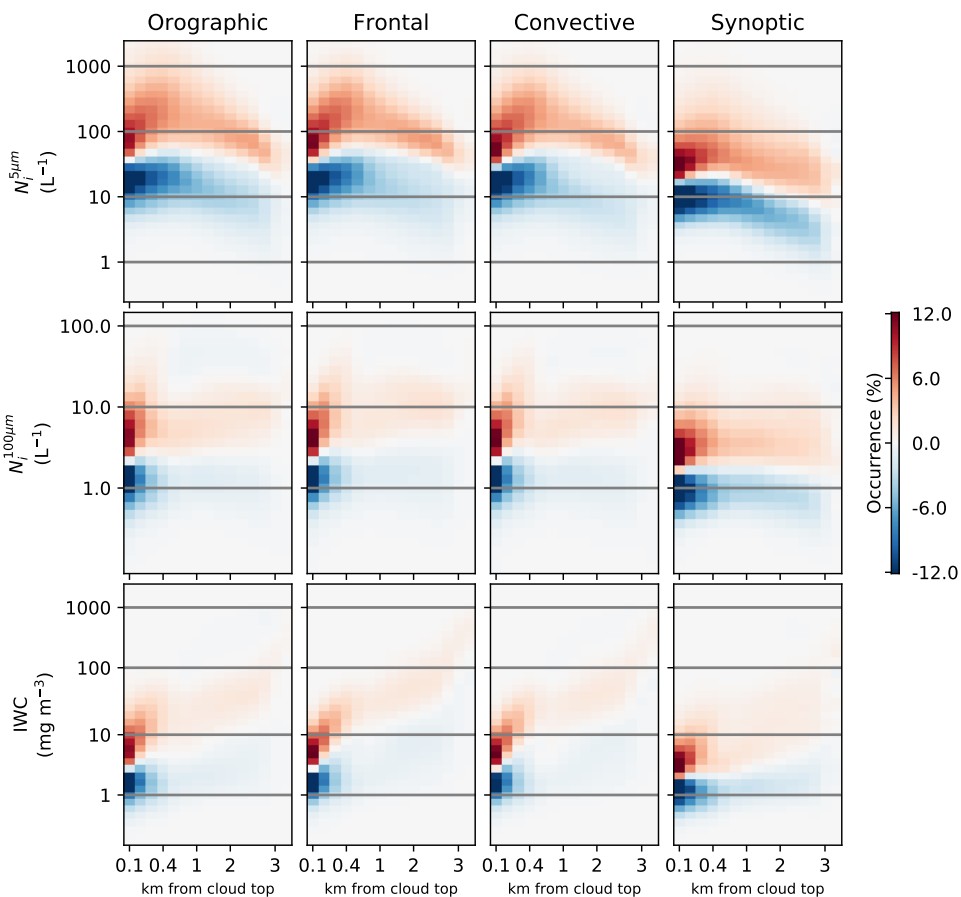

**Figure 9.** As Fig. 5, but showing the difference in the retrieved properties depending on the cloud top properties. Red over blue indicates that clouds with above median properties at the cloud top ($N_{i(top)}^{5\mu m}$, $IWC_{(top)}$) have higher values of the retrieved properties at a specified depth from the cloud top. Note the non-linear scale on the horizontal axis.

The changes in $N_{i(top)}^{5\mu m}$ observed in the previous section have impacts throughout the depth of the cloud. Fig. 9 shows how $N_i$ and IWC information propagates vertically within a cloud. The cloud profiles are split into two categories, based on whether they have above or below median values of the cloud top properties ($N_{i(top)}$, $IWC_{(top)}$). The difference in the vertical structure of the clouds (in a similar manner to Fig. 5) is shown, with red over blue indicating that an increase in the retrieved quantity at a given distance from the cloud top for profiles that were above median in that property at the cloud top.

The top row of Fig. 9 shows that $N_i^{5\mu m}$ information propagates a significant distance through the cloud. Clouds with an increased $N_{i(top)}^{5\mu m}$ maintain a higher $N_i^{5\mu m}$ at distances at least 3km from the cloud top in all regimes. However, as shown in the

second row, vertical information about the $N_i^{100\mu m}$ does not propagate nearly as far through the cloud. The vertical propagation is the highest in the synoptic regime. The vertical propagation of IWC information is very similar to the $N_i^{100\mu m}$, with the relationship to the cloud top IWC being significantly reduced more than 500 m from the cloud top.

The large vertical propagation of the $N_i^{5\mu m}$ indicates that the changes in $N_i^{5\mu m}$ at the cloud top found in the previous section can have considerable impact at lower levels in the cloud. However, the lower vertical propagation of the information about the larger crystals ($N_i^{100\mu m}$, IWC) would support the suggestion that the growth of the ice crystals after nucleation is primarily controlled by meteorological factors that do not play a large role in the nucleation processes that control $N_i^{100\mu m}$. Note that the temperature of the cloud top and the distance from the cloud top can still play a large role in determining the $N_i$ (Figs. 4, 5).

## 6 Discussion

These results show that the $N_{i(top)}^{5\mu m}$ is strongly affected by several factors including temperature (Fig. 1), cloud type (Fig. 2) and updraught (Fig. 3) and that changes in the $N_{i(top)}^{5\mu m}$ can be maintained at large distances from the cloud top (Fig. 9). The dependence of the $N_{i(top)}^{5\mu m}$ on the in-cloud updraught and the relationship to reanalysis liquid aerosol (Fig. 6) at temperatures between -35 °C and -60 °C is consistent with an impact of homogeneous nucleation processes on the $N_{i(top)}^{5\mu m}$. This is supported by the relationship of the $N_{i(top)}^{5\mu m}$ to the INP proxy (Fig. 8), where a reduction in $N_{i(top)}^{5\mu m}$ with increasing INP could be indicative of an INP suppression of homogeneous nucleation (Kärcher and Lohmann, 2003). The relationship with INP is also consistent with heterogeneous nucleation having a strong role to play in determining the $N_{i(top)}^{5\mu m}$ in synoptic cirrus clouds at temperatures colder than -60 °C.

Uncertainties in the retrieval have been covered in part one of this work (Sourdeval et al., submitted). However, there are a few points to note with regards to the relationship of the $N_i$ to other cloud and meteorological properties. Although there is significant uncertainty in the $N_i$ retrieval, many of these uncertainties are random errors and not systematic functions of the meteorological properties investigated here. Even ice crystal shape, which can be a major issue in ice cloud retrievals, is a function of temperature (to first order) and so does not impact the majority of the results which are presented in this work stratified by temperature. The geographical variations in Fig. 1b show a similar pattern to those from (Mitchell et al., 2016, 2018), with high $N_{i(top)}$ observed in mountainous regions and a reduced $N_{i(top)}$ in the tropics. The similarity of the results from these two different retrieval products, each with a different physical basis supports the conclusions drawn from these datasets regarding the global distribution of $N_i$. There is also little evidence to suggest that there are large biases caused by the retrieval only being able to use one instrument (radar or lidar). Cases where only the lidar detects a cloud are often characterised by monomodal ice distributions, which are well represented by the Delanoë et al. (2005) parametrisation. As such, these cases are retrieved with similar accuracy to the full radar-lidar retrieval (Sourdeval et al., submitted).

The cloud phase classification is of critical importance to the warmer clouds included in this study and there is evidence of a misclassification of a small number of cases at temperatures warmer than -35 °C (Fig. 8). This can make it difficult to interpret results at these temperatures, so they are not a focus of this work. The change in phase of these clouds as a function of aerosol is likely to dominate the radiative response of clouds to aerosols at these temperatures.

There are a number of limitations of this study that could be addressed in future work. The lack of information about the location of INP is a serious issue when investigating the impact of aerosol on $N_i$. While the INP proxy in this work is able to provide a qualitative estimate of the role of INP in determining the $N_i$, for a quantitative estimate a better proxy or measure of the global INP concentration is required.

Additionally, the impact of meteorological covariations makes it difficult to assign causality to the aerosol-$N_{i(top)}$ relationships observed in Fig. 6. The lack of a complete picture of the atmosphere makes it difficult to directly control for meteorological variability. The causal link between aerosol and $N_{i(top)}$ is thought to be strong (e.g. Kärcher, 2002), but the lack of observations of in-cloud updraughts also limits how accurately the impact of aerosol on the $N_i$ can be determined. Although the cloud regimes used have some ability to constrain the cloud-scale updraught (Gryspeerdt et al., 2017b), the updraught is a critical component in determining the $N_i$ through it's influence on the supersaturation. The in-cloud updraught is assumed to be largely independent of the aerosol properties in this work, but it is possible that the reanalysis aerosol is related to the in-cloud updraught, such that more aerosol is transported vertically in conditions with high in-cloud updraughts. In this case, a positive correlation between the $N_i$ and MACC reanalysis aerosol could be generated. However, as MACC does not explicitly simulate in-cloud updraughts, the impact of this confounding issue is likely to be small.

It is also possible that using cloud glaciation as a proxy allows other meteorological covariations, which could generate apparent relationships between the "INP" and $N_i$. However, the in-cloud updraught is of a second order importance in determining the cloud-top phase compared to the INP concentration (Bühl et al., 2013). The inclusion of a "glide-slope" test when determining the INP proxy means that it is also unlikely that clouds are being glaciated but undetected ice falling from higher cloud layers. The separation into cloud regimes also limits the impact of these kind of meteorological covariations, which might be expected between different regimes, but would be weaker within them.

The behaviour of the $N_i$ retrieval in this work follows the expected behaviour of the $N_i$ determined in several previous studies based on satellite remote sensing, in-situ, theoretical and modelling results. This provides further evidence that the DARDAR-LIM $N_i$ retrieval described in (Sourdeval et al., submitted) is able to retrieve the $N_i$ in a variety of situations.

## 7   Conclusions

Few global studies exist of the controls on the ice crystal number concentration ($N_i$), especially the role of aerosols. In this study, the DARDAR-LIM $N_i$ retrieval from part one (Sourdeval et al., submitted) is used to investigate possible controls on the $N_i$ at a global scale for the period 2006-2013. A special emphasis is placed on the $N_i$ at the cloud top ($N_{i(top)}$), due to the close proximity to ice crystal nucleation locations within many high clouds (Spichtinger and Gierens, 2009a; Diao et al., 2017).

Strong relationships between the $N_{i(top)}$ and updraught, cloud type and particularly temperature are observed (Figs. 1, 2), with a higher $N_{i(top)}$ for crystals larger then $5\mu m$ ($N_{i(top)}^{5\mu m}$) being found at colder temperatures in all regimes, consistent with an increased nucleation rate at lower temperatures. Fewer crystals larger than $100\mu m$, ($N_{i(top)}^{100\mu m}$) are found at the coldest temperatures, possibly due to the reduced depositional growth rate meaning that they sediment from the cloud top region before they can grow to a sufficient size.

Many of the regimes show an increase in the $N_i^{100\mu m}$ and a decrease in the $N_i^{5\mu m}$ with increasing distance from the cloud top (Fig. 5) due to the size sorting impact of sedimentation. The rate of change of the $N_i$ moving away from the cloud top depends on the regime, with much slower changes in the synoptic regime indicating a role of meteorological factors in determining ice crystal growth rates. This is supported by the weaker temperature dependence of the $N_i$ within clouds compared to the $N_{i(top)}$ (Fig. 4), which may also explain the apparent weak dependence of $N_i$ on temperature (Krämer et al., 2009) and INP (Kanji et al., 2017) found in previous studies. Given the large difference between the $N_{i(top)}$ and the $N_i$ deeper in the cloud, this may suggest that the cloud top would make a good target for future in-situ campaigns examining the controls on ice nucleation.

There are indications of homogeneous nucleation or possibly the freezing of liquid droplets determining the $N_{i(top)}$. At temperatures just colder than -35°C, there is a peak in the upper quantiles of the $N_{i(top)}^{5\mu m}$ (Fig. 2). This peak is related to the updraught strength in the cloud, with the reliably high updraughts in the orographic regime giving it the strongest peak (Fig. 3). This is further supported by the relationship between the $N_{i(top)}^{5\mu m}$ and the MACC reanalysis aerosol (Fig. 6), with an increased $N_{i(top)}^{5\mu m}$ being observed in high aerosol environments, indicating a possible dependence on the liquid aerosol concentration, particularly for smaller crystals, although this analysis cannot make a conclusive statement about the causality of this relationship. An investigation into the covariances between the MACC reanalysis aerosol, the DARDAR-LIM $N_i$ and meteorological factors is an important target for future work.

As previous work has suggested that INP occurrence is related to cloud glaciation, the glaciated fraction at -20° is used as a qualitative proxy for INP occurrence (Fig. 7). At temperatures between -50°C and -35°C, there is a reduction in $N_{i(top)}^{5\mu m}$ with increasing "INP" (Fig. 8), which may indicate an INP suppression of the homogeneous nucleation (Kärcher and Lohmann, 2003), providing further supporting evidence for the role of homogeneous nucleation in determining the $N_{i(top)}$. At colder temperatures, some regimes show an increasing $N_{i(top)}$, and the $N_{i(top)}^{100\mu m}$ in particular, which may be evidence of heterogeneous nucleation controlling the $N_{i(top)}$ and shifting the size distribution towards larger crystals. However, as with the relationship to liquid aerosol, meteorological covariations could be generating these relationships. Further studies are required to separate the role of these different mechanisms in controlling the $N_{i(top)}$ and to isolate the role of aerosols in these relationships.

While changes to the $N_{i(top)}$ are important for radiative considerations, changes in the $N_{i(top)}$ can have implications for the cloud many kilometres below the cloud top (Fig. 9). This far reaching impact into the lifecycle of ice and mixed-phase clouds demonstrates the importance of developing strong observational constraints on the controlling factors of the $N_i$. The results presented in this work provide a global context for existing theory and in-situ measurement based hypotheses about cloud properties, highlighting areas for future research to further constrain ice and mixed-phase cloud processes.

*Acknowledgements.* The MODIS data are from the NASA Goddard Space Flight Center (ftp://laadsweb.nascom.nasa.gov). The DARDAR data product were retrieved from the ICARE data center (http://www.icare.univ-lille1.fr). The MACC reanalysis data is available on-line at (http://apps.ecmwf.int/datasets/data/macc-reanalysis). This work was supported by funding from the European Research Council under the European Union's Seventh Framework Programme (FP7/2007-2013) / ERC grant agreement no. FP7-306284 ("QUAERERE"); the

Bundesministerium für Bildung und Forschung, grant numbers 01LK1210D, 01LK1503A and 01LK1505E; and Deutsche Forschungsge-meinschaft, grant number QU 311/14-1. EG is supported by an Imperial College Junior Research Fellowship.

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
