# Peer review of "Ice crystal number concentration estimates from lidar-radar satellite remote sensing. Part 2: Controls on the ice crystal number concentration"

_Atmospheric Chemistry and Physics, 2018_

## Referee Comment (RC1) · Anonymous Referee #1 · 3 Mar 2018

General Comments: This manuscript may advance our understanding of cirrus clouds considerably, especially in terms of homo- and heterogeneous ice nucleation and the dependence of those processes on topography, ice nuclei concentration, and aerosol concentration. However, the temperature dependence of the retrieved ice particle number concentration Ni (Dmin = 5 $\mu$m) appears at variance with global in situ observations of Ni, and this should be mentioned. Some recent literature was overlooked, and by discussing the results from these other studies, the arguments made in this study will be stronger. The manuscript is well organized and well written, and the quality of the

figures is good. Major and minor comments are listed below.

Major Comments:

1. Page 5, lines 11-19: Over what temperature domain is the "glaciation index" proxy for INP applied?

2. Figure 1a: These results make sense theoretically since homogeneous ice nucleation (hom) is sensitive to temperature. But in situ measurements in the tropical tropopause layer (TTL) show relatively low ice particle number concentrations (Ni, Dmin = 5 $\mu$m) there (e.g. Jensen et al., 2013, PNAS). Spichtinger and Krämer (2013, ACP) have offered a dynamical explanation for the relatively low TTL Ni (Ni < 30 L-1 typically). Since TTL cirrus appear more extensive and generally at higher altitudes than tropical anvil cirrus (Gasparini et al., 2017, J. Climate), it seems that Ni retrieved over the tropics at $\sim$ -70°C would be strongly influenced by TTL cirrus, but in Fig. 1a tropical Ni values are maximum at -70°C, being $\sim$ 135 L-1. Please discuss this apparent paradox.

3. Figure 1b and p. 6, lines 5-15: Fig. 1b is very similar to Figs. 11 and 12 in Mitchell et al. (2016, ACPD). Regarding the higher Ni over mountainous terrain outside the tropics, this finding and explanation was also reported in Mitchell et al. (2016). Although this paper was rejected since the editor felt the retrieved Ni values were too high, and therefore could not be used to infer nucleation modes, no arguments cast doubt on the spatial and temporal relative differences in Ni, which still appear meaningful. The results in Fig. 1b are more compelling when it is shown that two very different satellite retrieval techniques produce similar results in terms of the relative differences in Ni.

4. Page 7, lines 7-8: Please note here that the study by Krämer et al. (2009, ACP), based on five cirrus cloud field campaigns that measured Ni, does not show a strong temperature dependence for Ni. On average, Ni slightly decreases with decreasing temperature.

5. Page 7, lines 10-11: Perhaps I missed something, but I am not seeing Ni as high as 100 L-1 in Fig. 2 for T $\sim$ -15°C for the orographic and convective regimes.

6. Page 9, lines 32-34 and page 10, line 1: The Ni measurements reported in Krämer et al. (2009) were sampled over the size range 3.0–30 $\mu$m or 0.6–40 $\mu$m diameter, which accounted for at least 80% (but typically > 90%) of the total N in a PSD. Thus, these observations can be compared with Ni(Dmin = 5 $\mu$m) but not with Ni(Dmin = 100 $\mu$m).

7. Page 12, lines 6-14: Most of this argument is not clear to me, and moreover, the physics of cirrus clouds is very complex and does not lend itself to these simple arguments. The authors are encouraged to read Spichtinger and Gierens, Part 1a and 1b (2009, ACP).

8. Page 12, lines 29-32: Consider citing Zhao et al. (2018, ACP), since they use satellite remote sensing and cloud modeling to demonstrate how increasing aerosol concentrations act through homogeneous ice nucleation to decrease the effective radius in cirrus clouds (note that decreasing re often corresponds with increasing Ni).

9. Page 15, lines 12-13: Does not a higher INP concentration promote a LOWER supercooled liquid fraction over Siberia?

10. Page 15, lines 29-32: The study by Zhao et al. (2018, ACP) may be of interest, since they demonstrate that the relationship between cirrus cloud effective radius (re) and column aerosol optical depth (column AOD) and the relationship between re and the cirrus cloud layer dust AOD are similar. That is, for the region and time of study, there was a correlation between dust aerosols affecting cirrus clouds and the atmospheric column integrated AOD.

11. Page 16, lines 10-14: The "negative Twomey effect" described here was also observed in the satellite remote sensing study by Zhao et al. (2018, ACP).

12. Page 17, lines 3-9: It should be noted here that this argument assumes relatively

glaciated conditions at -20 C are indicative of relatively high INP concentrations for T < -50 C, which is stretching this assumption quite far.

Minor Comments: 1. Page 8, lines 4-5: By "increased homogeneous nucleation directly into the ice phase", are you referring to the freezing of aqueous haze aerosol particles?

2. Page 9, lines 2-3: Note that CCN do not need to be activated (i.e. cloud droplets) for homogeneous freezing; they can be dissolved as unactivated haze droplets (Koop et al., 2000, Nature). Perhaps this was the intention of this sentence, but it was not clear.

3. Page 9, lines 6-7: Barahona and Nenes (2008, JGR) are another good reference for demonstrating "the updraught limited nature of many cirrus clouds" regarding homogeneous ice nucleation.

4. Page 12, line4: Should "part one" be "Part 1"?

5. Figure 7: The "b" label is missing on this figure.

6. Page 19, line 23: Suggest modifying sentence to read: studies based on satellite remote sensing, in situ, theoretical and modeling results.

7. Page 19, lines 28-29: Good citations for this sentence are Diao et al. (2017, JGR), showing observational evidence for ice nucleation near cloud top, and Spichtinger and Gierens (2009, ACP), showing modeling evidence for this, and how nucleation rate profiles vary with updraft speed.

8. Page 20, line 11: A => At?

---

## Referee Comment (RC2) · Anonymous Referee #2 · 19 Mar 2018

**Review of "Ice crystal number concentration estimates from lidar-radar satellite retrievals. Part 2: Controls on the ice crystal number concentration" by E. Gryspeerdt et al.**

This paper uses the ice concentration retrievals described in Part 1 of the 2-part paper to investigate the relationships between ice concentration and both meteorological variables and aerosol properties. As described below, I have serious concerns with the paper as it is written, I do not think all of the conclusions are justified by the analysis presented, and I believe major revisions are required.

[Figure]

**1. Citations:** Examples where appropriate citations are omitted abound throughout the paper. Perhaps the authors are not familiar with the literature regarding cirrus ice concentrations, in which case I suggest the authors do a thorough literature search and cite appropriate papers. A few examples of missing references are provided here:

1. Page 1, lines 17-18: Numerous observational and modeling papers have been written by U.S. scientists on the issue of aerosol impacts on liquid clouds, but only European studies are cited here (two by co-authors of this paper!).

2. Page 1, line 19: Regarding the impact of aerosols on high clouds, again only one European paper has been cited, but there are many appropriate U.S. scientist-led papers (e.g., Jensen et al., 2016, JAS; Gettelman et al. papers; J. Penner group papers; etc.).

3. Page 2, line 8: In addition to the Korolev reference earlier papers (McFarquhar et al., 2007, GRL; Jensen et al., 2009, ACP) should be cited.

4. Page 2, lines 15-16: The climatology of ice concentrations published by Krämer et al. (2009) was based on a very limited set of measurements, particularly at low temperatures. Measurements from the ATTREX campaign near the tropical tropopause showed much higher ice concentrations at very cold temperatures (Jensen et al., 2013, PNAS; Jensen et al., 2016, JAS). Likewise, the ice concentrations shown by Muhlbauer et al. (2014) were perhaps biased by the limited sampling from a single campaign. The ice concentrations measured during MACPEX with a similar amount of data in the same geographical region and time of year showed the opposite trend with temperature (Jensen et al., 2013, JGR).

5. Page 2, lines 33-34: The dominance of dynamics has been pointed out in numerous other modeling studies (e.g., Jensen et al., 2013, JGR; Jensen et al., 2016, JAS; DeMott et al., 1997, JGR).

6. Page 3, line 8: DeMott et al. (1997, JGR) should also be cited here.

7. Page 3, lines 13-14: Jensen et al. (2016, JAS) should also be cited here.

8. Page 3, line 25: Again, McFarquhar et al. (2007, GRL) and Jensen et al. (2009, ACP) should be cited here.

9. Page 6, line 7: Barahona et al. (2017, Nature) should also be cited here.

10. Page 8, line 12: A number of earlier papers showed high ice concentrations in wave clouds (e.g., Jensen et al., 1998, GRL; Baker and Lawson, 2006, JAS).

**2. Page 3, lines 18-22:** Small-scale wave-driven vertical motions actually have been characterized by a number of aircraft measurements and super-pressure balloon measurements (e.g., Podglajen et al., 2016, GRL; Podglajen et al., 2017, JAS).

**3. Page 3, lines 27-28:** As noted in at least one of the referee comments on the Sourdeval et al. Part 1. paper, the ice concentration retrievals in regions with only lidar or only radar signals are highly suspect and insufficiently evaluated by comparison with in situ observations. This issue calls into question the results from this paper (Part 2.).

**4. Page 4, lines 11-13:** The cloud-top region only represents the conditions near nucleation zones during a short time period just after the transient, localized nucleation events. Differential sedimentation and entrainment rapidly reduce ice concentrations thereafter (e.g., Jensen et al., 2013, JGR). Only in wave clouds is it possible to know where the nucleation zone is located.

**5. Page 4, lines 20-22:** The classification scheme relies on MODIS data at 13:30 local solar time. So, are the ice concentrations in the remainder of the analysis restricted to times near this local time?

**6. Page 4, lines 32-33:** Actually abundant recent laboratory experiments have shown that organic-containing aerosols (which are abundant in the upper troposphere) will

likely be in a glassy state at low temperatures (e.g., Murray, 2008, ACP; Zobrist et al., 2008, J. Phys Chem.). The aerosols will likely be in a glassy state both at midlatitude and tropical upper troposphere conditions (Wilson et al., 2012, ACP).

**7. Page 5, lines 29-30:** Ice concentrations produced by heterogeneous nucleation should also increase with decreasing temperature. As shown by numerous laboratory and field studies, ice nuclei abundance increases with decreasing temperature (e.g., DeMott et al., 2010, PNAS).

**8. Figures 2 and 3, and discussion thereof:** It would be very helpful to provide a brief review of the regime classification from Gryspeerdt et al. (2017). In fact, it is impossible to evaluate the results shown in Figure 3 without knowing the different definitions of ORO 1 and ORO 2.

**9. Figure 2 discussion:** The differences in $N_i$ frequency distributions between different regimes are actually very slight, and I think they are somewhat exaggerated in the discussion. The main feature apparent in Figure 2 is a clear temperature dependence that is nearly identical in all of the regimes.

**10. Figure 2 discussion:** The text discusses a peak at temperatures just colder than -35°C. This peak is very subtle in most of the regimes, and in fact it occurs closer to -50°C. Therefore, I do not believe the assertion that there is a clear transition at the liquid water homogeneous freezing temperature (about -38°) is justified. If anything, such a transition would be smeared toward warmer temperatures by ice crystal sedimentation rather than shifted toward colder temperatures as apparent in Figure 2.

**11. Figure 3:** What are the units for the change in occurrence?

**12. Section 4.1:** As shown by previous modeling studies (e.g., Kärcher and Lohmann, 2002), the sensitivity of ice concentrations produced by homogeneous freezing of aqueous aerosol to aerosol abundance should be weak. Furthermore, it is entirely possible that $N_i$ frequency distribution differences shown in Figure 6 with different aerosol

loadings are simply a result of co-varying meteorology. For example, aerosol loading and ice concentrations could both be enhanced in regions with relatively strong mesoscale updrafts. In recent years, de-convolving the effects of co-varying meteorology has become a requirement for attributing changes in cloud properties to variations in aerosol properties. The same standard should be applied here. Either compelling evidence should be provided showing that the apparent changes in $N_i$ with aerosol loading are not caused by co-varying meteorology or the entire discussion in section 4.2 should be removed.

**13. Page 15, lines 14-19:** This paragraph starts by stating there is a strong correlation between occurrence of supercooled liquid and the mass concentration of reanalysis dust, then qualifications to this statement are made to acknowledge the lack of correlation in some regions. It would be clearer (and less misleading) to just state that correlations are only apparent in some regions.

**14. Section 4.2.2** The same issue about co-varying meteorology discussed above for section 4.1 applies here. Again, either a clear demonstration that co-varying meteorology is not the cause of the correlations needs to be provided, or the section should be removed.

**15. Section 5** The same issue about co-varying meteorology discussed above for section 4.1 applies here. Again, either a clear demonstration that co-varying meteorology is no the cause of the correlations needs to be provided, or the section should be removed.

**16. Discussion and Conclusions sections:** The authors should remove unjustified conclusions. See comments above regarding a transition at the homogeneous freezing threshold, dust impacts, and co-varying meteorology.

---

## Author Comment (AC1) · 17 Jul 2018

**Response to reviewer 1**

**General Comments**
* * *
*: This manuscript may advance our understanding of cirrus clouds considerably, especially in terms of homo- and heterogeneous ice nucleation and the dependence of those processes on topography, ice nuclei concentration, and aerosol concentration. However, the temperature dependence of the retrieved ice particle number concentration Ni (Dmin = 5m) appears at variance with global in situ observations of Ni, and this should be mentioned. Some recent literature was overlooked, and by discussing the results from these other studies, the arguments made in this study will be stronger. The manuscript is well organized and well written, and the quality of the figures is good. Major and minor comments are listed below.*

**Reply**: We thank the reviewer for their useful comments and address them in turn below. The temperature dependence of $N_i^{5\mu m}$ has been covered in more detail in the response to part 1, but it is also noted briefly in this work. When compared to the temperature dependence in Krämer et al (2009), there is indeed a strong difference between the in-situ and satellite measurements. However, as shown in the response to reviewer two of part 1, the DARDAR-LIM methodology would be able to reproduce the weak temperature dependence if it was confined to the same locations. It should also be noted that many previous studies do not target the cloud top, resulting in a weaker temperature dependence and that the Krämer study also included some cirrus formed over orography. This would make Fig. 4a the closest comparison, which has a weak (although non-zero) temperature dependence at temperatures colder than -35C.

**Major comments**
* * *
**Page 5, lines 11-19**: *Over what temperature domain is the "glaciation index" proxy for INP applied?*

**Reply**: The glaciation index is determined using all temperatures, but in practice, only temperatures between 0C and about -40C contribute to the index. As ice is not found at temperatures warmer than 0C, "warm ice" can only occur between 0C and -20C. Similarly "cold liquid" is only found between -20C and -40C.

The proxy itself is applied at temperatures colder than -35°C. We agree that this is perhaps treating the proxy in a to idealised fashion, but as stated in section 4.2, the worst case scenario is that there is no correlation between INP and the proxy, in which case there should be no relationship between the proxy and Ni. The method section has been expanded upon to make this clearer and the discussion around this has been modified to point out the possible important role of meteorological covariations.
* * *
**Figure 1a**: *These results make sense theoretically since homogeneous ice nucleation (hom) is sensitive to temperature. But in situ measurements in the*

*tropical tropopause layer (TTL) show relatively low ice particle number concentrations (Ni, Dmin= 5m) there (e.g. Jensen et al., 2013, PNAS). Spichtinger and Krämer (2013, ACP) have offered a dynamical explanation for the relatively low TTL Ni (Ni < 30 L-1 typically). Since TTL cirrus appear more extensive and generally at higher altitudes than tropical anvil cirrus (Gasparini et al., 2017, J. Climate), it seems that Ni retrieved over the tropics at ≈-70C would be strongly influenced by TTL cirrus, but in Fig. 1a tropical Ni values are maximum at -70C, being ≈135 L-1. Please discuss this apparent paradox.*

**Reply**: Thank you for pointing this out. Part of the issue comes from the skewed Ni distribution, such that this simple average in the plot here is strongly controlled by the higher values of Ni. The joint histograms later in the paper provide a better description of the actual number concentration values (a sentence on this has now been included where the joint histograms are introduced).

The difference in the values reported in this work may also be due to sampling differences between aircraft and the satellite retrieval. As TTL cirrus can be very thin (under 200m thick), these may occasionally be missed by the DARDAR retrieval, particularly if they have a very low $N_i$ concentration. Similarly, it is possible that the sampling during aircraft campaigns is selective. Uncertainties in assumed parameters of the size distribution may also lead to an overestimation in $N_i$, as much as a factor of two, that could explain a large part of this discrepancy.

However, figures included in the response to Part 1 show that if the DARDAR retrieval is sampled in a similar manner to Krämer et al (2009), it produces a similar $N_i$ magnitude and temperature dependence. This would give some confidence in the results of the retrieval, suggesting that a sampling difference is the primary cause of the difference between the satellite and in-situ results. This is now further discussed in section 3.2
* * *
**Figure 1b and p. 6, lines 5-15**: *Fig. 1b is very similar to Figs. 11 and 12 in Mitchell et al. (2016, ACPD). Regarding the higher Ni over mountainous terrain outside the tropics, this finding and explanation was also reported in Mitchell et al. (2016). Although this paper was rejected since the editor felt the retrieved Ni values were too high, and therefore could not be used to infer nucleation modes, no arguments cast doubt on the spatial and temporal relative differences in Ni, which still appear meaningful. The results in Fig. 1b are more compelling when it is shown that two very different satellite retrieval techniques produce similar results in terms of the relative differences in Ni.*

**Reply**: There are a number of similarities, many thanks for pointing this out. This paper and the updated 2018 version have now been mentioned here and in a section in the discussion on the similarity between the two retrievals and the support this provides to conclusions drawn from both datasets.
* * *
**Page 7, lines 7-8**: *Please note here that the study by Krämer et al. (2009, ACP), based on five cirrus cloud field campaigns that measured Ni, does not show a strong temperature dependence for Ni. On average, Ni slightly decreases with decreasing temperature.*

**Reply**: We agree that this is a puzzling difference between the two different datasets. This has been covered in more detail in the response to part one. While the temperature dependence in this work is different to the slight weak temperature dependence shown in Krämer et al (2009), the temperature dependence of these results is significantly weaker if only regions colder than -40C are considered. At least part of the weak temperature dependence of Krämer et al (2009) is also due to the vertical distribution of $N_i$ within a cloud. The internal $N_{i(top)}$ plots in Fig. 4 are much closer to the temperature dependence from Krämer et al (2009), especially in this temperature range. There also appears to be a difference due to the different sampling of the satellite and aircraft measurements (see Part 1). The discussion on this at the end of the subsection has now been improved.
* * *
**Page 7, lines 10-11**: *Perhaps I missed something, but I am not seeing Ni as high as 100 L-1 in Fig. 2 for T≈-15C for the orographic and convective regimes.*
**Reply**: The plots have all been modified to make them clearer and more useful. As part of this, the lower frequency of occurrence of these retrievals is now more clearly visible. The number has also been reduced to a less sensational 50L-1.
* * *
**Page 9, lines 32-34 and page 10, line 1**: *The Ni measurements reported in Krämer et al. (2009) were sampled over the size range 3.0-30m or 0.6-40m diameter, which accounted for at least 80% (but typically > 90%) of the total N in a PSD. Thus, these observations can be compared with Ni(Dmin = 5m) but not with Ni(Dmin = 100m).*
**Reply**: A very good point. The enticing direct comparison to $N_i^{100\mu m}$ has been removed. This discussion has been improved, instead focussing on the weaker temperature dependence of the $N_{i(top)}^{5\mu m}$ at temperatures colder than -35C. Further information on this has been included in the response to Part 1.
* * *
**Page 12, lines 6-14**: *Most of this argument is not clear to me, and moreover, the physics of cirrus clouds is very complex and does not lend itself to these simple arguments. The authors are encouraged to read Spichtinger and Gierens, Part 1a and 1b (2009, ACP).*
**Reply**: We agree that this might have been a step too far. This section has now been amended to refer to the temperature and regime dependence of the peak as evidence for the impact of homogenous nucleation. The discussion of the peak providing information on updraught speed itself has been removed.

Further investigation showed that this peak is strongly temperature dependent , disappearing when clouds with colder tops are considered (plot included in supplement). Given the temperature dependence of $N_{i(top)}$, with an increase for temperatures colder than -35C, the impact of updraught on the $N_{i(top)}^{5\mu m}$ and the relationship between MACC aerosol and $N_{i(top)}^{5\mu m}$ are all suggestive of homogeneous nucleation. This peak would be consistent with a nucleation region below the cloud top. While this is not the case in all cloud types, the size of this peak region is broadly consistent with the thicknesses of the nucleation region

noted in Jensen et al. (2016), of around 20 to 500m.
* * *
**Page 12, lines 29-32**: *Consider citing Zhao et al. (2018, ACP), since they use satellite remote sensing and cloud modeling to demonstrate how increasing aerosol concentrations act through homogeneous ice nucleation to decrease the effective radius in cirrus clouds (note that decreasing re often corresponds with increasing Ni).*
**Reply**: Many thanks for pointing this out. This paper has been included, along with Jiang et al, ACP, 2011 and Chylek et al, GRL, 2006, which also provide evidence of a variation in crystal size with changing aerosol environments.
* * *
**Page 15, lines 12-13**: *Does not a higher INP concentration promote a LOWER supercooled liquid fraction over Siberia?*
**Reply**: This has been amended to point out that the supercooled fraction is lower than expected, given the lack of high level dust in this region.
* * *
**Page 15, lines 29-32**: *The study by Zhao et al. (2018, ACP) may be of interest, since they demonstrate that the relationship between cirrus cloud effective radius (re) and column aerosol optical depth (column AOD) and the relationship between re and the cirrus cloud layer dust AOD are similar. That is, for the region and time of study, there was a correlation between dust aerosols affecting cirrus clouds and the atmospheric column integrated AOD.*
**Reply**: Many thanks for pointing this out. A reference to this work has been included here and the discussion has been slightly expanded to include other studies of aerosol vertical autocorrelation.
* * *
**Page 16, lines 10-14**: *The "negative Twomey effect" described here was also observed in the satellite remote sensing study by Zhao et al. (2018, ACP).*
**Reply**: This is now noted.
* * *
**Page 17, lines 3-9**: *It should be noted here that this argument assumes relatively glaciated conditions at -20 C are indicative of relatively high INP concentrations for T <-50 C, which is stretching this assumption quite far.*
**Reply**: We agree that this is quite a stretch, but in the worst case this would produce no correlation between this INP proxy and the $N_i$. The end of this paragraph has now been modified to note this. There is some evidence of significant vertical autocorrelation in aerosol (Weigum, 2014; Stier, 2016), but the applicability of this proxy will be an area of future investigation.

**Minor Comments**
* * *
**Page 8, lines 4-5**: *By "increased homogeneous nucleation directly into the ice phase", are you referring to the freezing of aqueous haze aerosol particles?*
**Reply**: Amended
* * *
**Page 9, lines 2-3**: *Note that CCN do not need to be activated (i.e. cloud droplets) for homogeneous freezing; they can be dissolved as unactivated haze*

*droplets (Koop et al., 2000, Nature). Perhaps this was the intention of this sentence, but it was not clear.*
**Reply**: Thank you for pointing this out. The sentence has been amended
* * *
**Page 9, lines 6-7***: Barahona and Nenes (2008, JGR) are another good reference for demonstrating "the updraught limited nature of many cirrus clouds" regarding homogeneous ice nucleation.*
**Reply**: Included
* * *
**Page 12, line4***: Should "part one" be "Part 1"?*
**Reply**: Amended
* * *
**Figure 7***: The "b" label is missing on this figure.*
**Reply**: Amended
* * *
**Page 19, line 23***: Suggest modifying sentence to read: studies based on satellite remote sensing, in situ, theoretical and modeling results.*
**Reply**: Amended
* * *
**Page 19, lines 28-29***: Good citations for this sentence are Diao et al. (2017, JGR), showing observational evidence for ice nucleation near cloud top, and Spichtinger and Gierens (2009, ACP), showing modeling evidence for this, and how nucleation rate profiles vary with updraft speed.*
**Reply**: Amended
* * *
**Page 20, line 11***: A => At?*
**Reply**: Amended

**Response to reviewer 2**

**General Comments:**
* * *
*: This paper uses the ice concentration retrievals described in Part 1 of the 2-part paper to investigate the relationships between ice concentration and both meteorological variables and aerosol properties. As described below, I have serious concerns with the paper as it is written, I do not think all of the conclusions are justified by the analysis presented, and I believe major revisions are required.*
**Reply**: We thank the reviewer for their useful comments and address them in turn below
* * *
**1. Citations**: *Examples where appropriate citations are omitted abound throughout the paper. Perhaps the authors are not familiar with the literature regarding cirrus ice concentrations, in which case I suggest the authors do a thorough literature search and cite appropriate papers. A few examples of missing references are provided here:*

- *Page 1, lines 17-18: Numerous observational and modeling papers have been written by U.S. scientists on the issue of aerosol impacts on liquid clouds, but only European studies are cited here (two by co-authors of this paper!).*

- *Page 1, line 19: Regarding the impact of aerosols on high clouds, again only one European paper has been cited, but there are many appropriate U.S. scientist-led papers (e.g., Jensen et al., 2016, JAS; Gettelman et al. papers; J. Penner group papers; etc.).*

- *Page 2, line 8: In addition to the Korolev reference earlier papers (McFarquhar et al., 2007, GRL; Jensen et al., 2009, ACP) should be cited.*

- *Page 2, lines 15-16: The climatology of ice concentrations published by Krämer et al. (2009) was based on a very limited set of measurements, particularly at low temperatures. Measurements from the ATTREX campaign near the tropical tropopause showed much higher ice concentrations at very cold temperatures (Jensen et al., 2013, PNAS; Jensen et al., 2016, JAS). Likewise, the ice concentrations shown by Muhlbauer et al. (2014) were perhaps biased by the limited sampling from a single campaign. The ice concentrations measured during MACPEX with a similar amount of data in the same geographical region and time of year showed the opposite trend with temperature (Jensen et al., 2013, JGR).*

- *Page 2, lines 33-34: The dominance of dynamics has been pointed out in numer- ous other modeling studies (e.g., Jensen et al., 2013, JGR; Jensen et al., 2016, JAS; DeMott et al., 1997, JGR).*

- *Page 3, line 8: DeMott et al. (1997, JGR) should also be cited here.*

- *Page 3, lines 13-14: Jensen et al. (2016, JAS) should also be cited here.*

- *Page 3, line 25: Again, McFarquhar et al. (2007, GRL) and Jensen et al. (2009, ACP) should be cited here.*

- *Page 6, line 7: Barahona et al. (2017, Nature) should also be cited here.*

- *Page 8, line 12: A number of earlier papers showed high ice concentrations in wave clouds (e.g., Jensen et al., 1998, GRL; Baker and Lawson, 2006, JAS).*

**Reply**: Many thanks for pointing this out. The oversight of U.S. scientists was not intentional, the papers picked in the brief section on liquid clouds were intended to be only a selection of papers that highlighted the central role of $N_d$ for generating observational constraints of the aerosol impact on liquid clouds. This was a significant oversight which has hopefully now been remedied to better indicate the international composition of this field. Many thanks for the suggested omitted references, these have now been included in the revised version.
* * *
**Page 3, lines 18-22**: *Small-scale wave-driven vertical motions actually have been characterized by a number of aircraft measurements and super-pressure balloon measurements (e.g., Podglajen et al., 2016, GRL; Podglajen et al., 2017, JAS)*
**Reply**: Included
* * *
**Page 3, lines 27-28**: *As noted in at least one of the referee comments on the Sourdeval et al. Part 1. paper, the ice concentration retrievals in regions with only lidar or only radar signals are highly suspect and insufficiently evaluated by comparison with in situ observations. This issue calls into question the results from this paper (Part 2.).*
**Reply**: This has been primarily addressed in the comments for Part 1. In summary, there is less information available in the lidar only or radar only portions of the cloud such that these retrievals would have a higher error. However, this does not translate into a strong bias against the in-situ measurements, although there is still significant scatter in the comparison.

Part of the weak impact of the lidar-only regions on the overall statistics comes from the clouds where only a lidar signal is received. These clouds tend to have a monomodal size distribution, such that although the large crystals are poorly characterised by the lidar, they are not present in significant enough numbers to bias the $N_i$ retrieval. Surprisingly, the retrievals in the lidar only-portions of the cloud may even be more accurate due to the simpler shape of the $N_i$ size distribution.

The impact of a transition between lidar-only and lidar-radar data is expanded in the section on the vertical distribution of $N_i$, as this is the section where it would play the clearest role, although this is also now referenced in the discussion.
* * *
**Page 4, lines 11-13**: *The cloud-top region only represents the conditions near nucleation zones during a short time period just after the transient, localized nucleation events. Differential sedimentation and entrainment rapidly reduce ice*

*concentrations thereafter (e.g., Jensen et al., 2013, JGR). Only in wave clouds is it possible to know where the nucleation zone is located.*

**Reply**: This is a good point, we have modified the description here to note that although the nucleation region is sometime located at cloud top, this is not always the case. However, as the coldest part of the cloud, the cloud top is the location of the theoretical maximum nucleation rate and so is a useful method for reducing the impact of temperature variability due to the vertical extent of the cloud. As is noted in the vertical $N_i$ structure section, the highest $N_i$ values are rarely found directly at the cloud top, but they are usually located within 500m of the cloud top.
* * *
**Page 4, lines 20-22**: *The classification scheme relies on MODIS data at 13:30 local solar time. So, are the ice concentrations in the remainder of the analysis restricted to times near this local time?*

**Reply**: This analysis uses only daytime data, which restricts it to the 13:30 LST time of the classification scheme through the orbit of the satellites used to construct the DARDAR-LIM dataset. This is now mentioned in the methods section.
* * *
**Page 4, lines 32-33**: *Actually abundant recent laboratory experiments have shown that organic-containing aerosols (which are abundant in the upper troposphere) will likely be in a glassy state at low temperatures (e.g., Murray, 2008, ACP; Zobrist et al., 2008, J. Phys Chem.). The aerosols will likely be in a glassy state both at midlatitude and tropical upper troposphere conditions (Wilson et al., 2012, ACP).*

**Reply**: Thank you for pointing this out. A note on glassy organic aerosol has now been included in this section of the methods and is discussed further in the section on liquid aerosol and $N_{i(top)}$.
* * *
**Page 5, lines 29-30**: *Ice concentrations produced by heterogeneous nucleation should also increase with decreasing temperature. As shown by numerous laboratory and field studies, ice nuclei abundance increases with decreasing temperature (e.g., DeMott et al., 2010, PNAS).*

**Reply**: A note to this effect has been included in this paragraph. In addition, the gridline in the temperature-number histograms have been modified to show the number of INP from the Demott (2010) parametrisation as a function of temperature to further illustrate this effect.
* * *
**Figures 2 and 3, and discussion thereof**: *It would be very helpful to provide a brief review of the regime classification from Gryspeerdt et al. (2017). In fact, it is impossible to evaluate the results shown in Figure 3 without knowing the different definitions of ORO 1 and ORO 2.*

**Reply**: Thank you for pointing this out. An brief explanation of the regimes has now been included in the methods section.
* * *
**Figure 2 discussion**: *The differences in Ni frequency distributions between different regimes are actually very slight, and I think they are somewhat exaggerated*

*in the discussion. The main feature apparent in Figure 2 is a clear temperature dependence that is nearly identical in all of the regimes.*

**Reply**: We agree that the differences between the regimes are smaller than the temperature dependence within them. However, there are strong differences between the regimes, as shown by Fig. 3. A separate subplot has been added to Fig. 3, showing the difference between the frontal and synoptic regimes, an even larger change than found amongst the orographic regimes. While this is less clearly due to updraught, it is likely related, demonstrating the important differences between the regimes.
* * *
**Figure 2 discussion***: The text discusses a peak at temperatures just colder than -35C. This peak is very subtle in most of the regimes, and in fact it occurs closer to -50C. Therefore, I do not believe the assertion that there is a clear transition at the liquid water homogeneous freezing temperature (about -38) is justified. If anything, such a transition would be smeared toward warmer temperatures by ice crystal sedimentation rather than shifted toward colder temperatures as apparent in Figure 2.*

**Reply**: We agree that the effect is very subtle in some of the regimes, but we consider this supporting evidence for the impact of homogeneous freezing, as the transition is stronger in regimes with a higher expected updraught (Fig. 3).

By using the cloud top temperature and $N_i$, this means that the temperature given is the coldest temperature in that cloud. As such, that transition cannot be smeared to warmer temperatures, as these clouds cannot achieve the temperatures necessary for homogenous freezing of liquid droplets or haze. However, even if this was an effect that took place purely at -38C, it would likely still be visible in clouds with tops significantly colder than this temperature as updraughts carry the tops higher. This would result in the effect being smeared to colder temperatures, as is observed in Fig. 2.

The weak difference between the Oro2 and Oro1 regimes (expected to differ primarily by updraught environment) warmer than -35C is another indicator of the role of homogenous processes (Fig. 3). These is no process in the retrieval that changes at this temperature other than the phase classification. We therefore feel that the clear transition in both panels of this figure is a strong indicator of homogenous freezing of either liquid droplets or haze.
* * *
**Figure 3:***: What are the units for the change in occurrence?*

**Reply**: Change in absolute percentage occurrence. This has now been added to the figure
* * *
**Section 4.1***: As shown by previous modeling studies (e.g., Kärcher and Lohmann, 2002), the sensitivity of ice concentrations produced by homogeneous freezing of aqueous aerosol to aerosol abundance should be weak. Furthermore, it is entirely possible that Ni frequency distribution differences shown in Figure 6 with different aerosol loadings are simply a result of co-varying meteorology. For example, aerosol load- ing and ice concentrations could both be enhanced in regions with relatively strong mesoscale updrafts. In recent years, de-convolving the effects of co-varying meteorology has become a requirement for attributing changes in*

*cloud properties to variations in aerosol properties. The same standard should be applied here. Either compelling evidence should be provided showing that the apparent changes in Ni with aerosol loading are not caused by co-varying meteorology or the entire discussion in section 4.2 should be removed.*

**Reply**: The results shown here do not differ strongly from the changes proposed by Kärcher and Lohmann. Although there is a clear relationship to MACC aerosol in Fig. 6, the actual change in $N_i$ this implies is relatively small, especially for such a large aerosol perturbation. The mean $N_{i(top)}$ values are shown in the attached plot, but it is clear to see a strong increase in $N_{i(top)}^{5\mu m}$ for the orographic regime at about -38C. The difference related to the MACC aerosol is only visible at temperatures warmer than this, but there is around a 25% increase in the $N_{i(top)}$ for a large change in aerosol. At around -50C, this would require an updraught larger than around 1ms-1. This is large, but still plausible for the orographic and convective regimes (Gryspeerdt et al., 2017).

[Figure]

While these results demonstrate plausible properties and have a sensible magnitude for an aerosol effect, we are very aware that this is only an observed relationship and that we have not been able to demonstrate causality. Work is currently underway to investigate this, but conclusively showing this in an observational study such as this is near impossible. As such, the section on meteorological covariations in the discussion has been highlighted and it is now mentioned in the results section. However, these results should not be removed from this work purely because they have not met the incredibly high bar of proving causality. It is useful to know that the $N_{i(top)}^{5\mu m}$ and reanalysis aerosol are correlated and as mentioned in the discussion section, there is reasonable cause to believe that in-cloud updraughts are not the cause of the observed aerosol-$N_{i(top)}^{5\mu m}$ relationship as they are poorly simulated in MACC.
* * *
**Page 15, lines 14-19**: *This paragraph starts by stating there is a strong correlation between occurrence of supercooled liquid and the mass concentration of reanalysis dust, then qualifications to this statement are made to acknowledge the lack of correlation in some regions. It would be clearer (and less misleading) to just state that correlations are only apparent in some regions.*

**Reply**: This paragraph has now been re-worded to following this suggestion.

**Section 4.2.2**: *The same issue about co-varying meteorology discussed above for section 4.1 applies here. Again, either a clear demonstration that co-varying meteorology is not the cause of the correlations needs to be provided, or the section should be removed.*

**Reply**: Following the above discussion, we have amended to discussion to make it clear that co-variation could be the cause for this relationship, rewording the second half of this section.

**Section 5**: *The same issue about co-varying meteorology discussed above for section 4.1 applies here. Again, either a clear demonstration that co-varying meteorology is no the cause of the correlations needs to be provided, or the section should be removed.*

**Reply**: Following the above discussion, we have modified some of the discussion about the possible impact of meteorological covariations, but we do not feel that it is necessary to remove this section for the reasons outlined above.

**Discussion and Conclusions sections**: *The authors should remove unjustified conclusions. See comments above regarding a transition at the homogeneous freezing threshold, dust impacts, and co-varying meteorology.*

**Reply**: Thanks to the reviewers comments, we believe we have better justified some of these conclusions. However, as they note, it is not possible to rule out the impact of meteorological covariations in a study such as this one. As such, we have included a better discussion on this impact and modified the conclusions in this area to highlight the need to clearly isolate the aerosol effects from possible meteorological covariations.

**Bibliography**

[revised manuscript text omitted]

---

## Referee Report (RR1)

**Second review of "Ice crystal number concentration estimates from lidar-radar satellite remote sensing. Part 2: Controls on the ice crystal number concentration" by E. Gryspeerdt et al.**

The authors have generally done a good job of responding to the comments from the first round of reviews and the revisions to the manuscript are generally appropriate. However, I still have a few concerns as described below.

**1. Response to comment and text revisions with regard to the peak in $N_{i(top)}^{5\mu m}$ at T just below -35C:** I still think this feature is somewhat exaggerated by the discussion in the text. Looking again at Figure 2, it seems to me that the the peak is only apparent in the upper envelope of the frequency distributions. No peak (or even change in slope) is visible in the most frequent (red) occurrences of $N_{i(top)}^{5\mu m}$ versus temperature. Also, I still think any feature associated with homogeneous freezing of pure droplets should be smeared downward by ice sedimentation rather than extending to lower temperatures (higher altitudes). Except in strong convective updrafts, the ice crystals are typically large enough to fall relative to the vertical uplift driving the cloud formation. The cloud top may rise with time as the updraft continues and ice nucleation propagates upward to lower temperatures, but this would not shift down the temperature at which there is a change in the mode of ice nucleation.

**2. Page 20, line 10:** It is plausible that an increased dominance of heterogeneous nucleation over homogeneous freezing can reduce ice concentrations. However, I think it is a bit misleading to call it a "negative Twomey effect". The ice concentration is not decreasing in response to the overall aerosol abundance. Rather it is a response to a change in the abundance of a small subset of the aerosol population.

**Discussion of co-varying meteorology:** The authors have added appropriate caveats about the possibility that the correlations between MACC aerosol product and retrieved ice concentrations may be a result of covariances between meteorology and aerosol processes. They suggest that investigation of such covariances is not possible with this analysis. I would think the extensive statistics available with the satellite retrievals would make such a study possible. I'm fine with the authors stating that such an analysis is beyond the scope of this paper. Perhaps they could suggest that the covariance analysis should be undertaken in a future study.

---

## Author Response (AR2)

**Response to reviewer**

1. Response to comment and text revisions with regard to the peak in  $N_{i(top)}^{5\mu m}$ at T just below -35C:: I still think this feature is somewhat exaggerated by the discussion in the text. Looking again at Figure 2, it seems to me that the the peak is only apparent in the upper envelope of the frequency distributions. No peak (or even change in slope) is visible in the most frequent (red) occurrences of N5m i(top) versus temperature. Also, I still think any feature associated with homogeneous freezing of pure droplets should be smeared downward by ice sedimentation rather than extending to lower temperatures (higher altitudes). Except in strong convective updrafts, the ice crystals are typically large enough to fall relative to the vertical uplift driving the cloud formation. The cloud top may rise with time as the updraft continues and ice nucleation propagates upward to lower temperatures, but this would not shift down the temperature at which there is a change in the mode of ice nucleation.

**Reply**: We have toned down comments on the strength of this peak, modifying the paragraph describing the peak to read

"All of the regimes also show a peak in the highest  $N_{i(top)}^{5\mu m}$  percentiles at temperatures just colder than -35°C. The strength varies by regime, with the orographic regime showing a stronger peak and only a weak peak being observed in the frontal and convective regimes. The peak is barely present in the synoptic regime, where the  $N_{i(top)}^{5\mu m}$  is located on the trend in  $N_{i(top)}^{5\mu m}$  present at other temperatures. An increase in the largest  $N_{i(top)}^{5\mu m}$  values at this temperature is consistent with homogeneous nucleation, either through an increase in the freezing of liquid droplets or by increased homogeneous nucleation through the freezing of unactivated aqueous haze particles. ..."

References to the "strong peak" are also changed to just "peak".

We agree that a feature associated with homogeneous nucleation would normally be smeared to warmer temperatures by sedimentation. However, this effect is not visible in Fig. 2 as it shows only the cloud top. Representing approximately the coldest temperature in the cloud, clouds with tops warmer than -35C cannot have any appreciable impact of homogeneous nucleation. The higher updraught speeds required to generate supersaturations for homogeneous nucleation would also suggest that descending cloud tops are unlikely to smear the peak to warmer temperatures. The only other way for the peak to be smeared to warmer temperatures is through an error in the reanalysis temperature, which explain why there is little evidence of the peak being smeared to warmer temperatures.

An updraught effect increasing the height of the cloud top would not affect the temperature at which homogeneous nucleation occurs, but it would smear the occurrence of a peak in the  $N_{i(top)}^{5\mu m}$  to colder temperatures by changing the height of cloud tops when homogeneous nucleation is occurring, moving them to these colder temperatures. This explains why the peak is very weak warmer than -35C, with the upper  $N_{i(top)}^{5\mu m}$  values increasing only at temperatures colder than -35C, but being smeared to colder temperatures.

2. Page 20, line 10: It is plausible that an increased dominance of heterogeneous nucleation over homogeneous freezing can reduce ice concentrations. However, I think it is a bit misleading to call it a "negative Twomey effect". The ice concentration is not decreasing in response to the overall aerosol abundance. Rather it is a response to a change in the abundance of a small subset of the aerosol population

**Reply**: We agree that it does not act exactly opposite to the impact of the standard Twomey effect in liquid clouds. The first reference to the "negative Twomey effect" has been removed. In the other two cases, the references to a "negative Twomey effect" have been replaced by a reference to an INP suppression of homogeneous nucleation.

**Discussion of co-varying meteorology::** The authors have added appropriate caveats about the possibility that the correlations between MACC aerosol product and retrieved ice concentrations may be a result of covariances between meteorology and aerosol processes. They suggest that investigation of such covariances is not possible with this analysis. I would think the extensive statistics available with the satellite retrievals would make such a study possible. I'm fine with the authors stating that such an analysis is beyond the scope of this paper. Perhaps they could suggest that the covariance analysis should be undertaken in a future study.

**Reply**: A sentence has been added to the end of the fourth paragraph of the conclusions, reading "An investigation into the covariances between the MACC reanalysis aerosol, the DARDAR-LIM  $N_i$  and meteorological factors is an important target for future work."

**Ice crystal number concentration estimates from lidar-radar satellite remote sensing. Part 2: Controls on the ice crystal number concentration**

Edward Gryspeerdt1, Odran Sourdeval2, Johannes Quaas2, Julien Delanoë3, Martina Krämer4, and Philipp Kühne2

1Space and Atmospheric Physics Group, Imperial College London, London, United Kingdom
 2Institute for Meteorology, Universität Leipzig, Germany
 3Laboratoire Atmosphères, Milieux, Observations Spatiales/IPSL/UVSQ/CNRS/UPMC, Guyancourt, France
 4Forschungszentrum Jülich, Institut für Energie und Klimaforschung (IEK-7), Jülich, Germany

**Correspondence:** Edward Gryspeerdt (e.gryspeerdt@imperial.ac.uk)

Abstract. The ice crystal number concentration  $(N_i)$  is a key property of ice clouds, both radiatively and microphysically. However, due to sparse in-situ measurements of ice cloud properties, the controls on the  $N_i$  have remained difficult to determine. As more advanced treatments of ice clouds are included in global models, it is becoming increasingly necessary to develop strong observational constraints on the processes involved.

This work uses the DARDAR-LIM  $N_i$  retrieval described in part one to investigate the controls of the  $N_i$  at a global scale. The retrieved clouds are separated by type. The effects of temperature, proxies for in-cloud updraught and aerosol concentrations are investigated. Variations in the cloud top  $N_i$  ( $N_{i(top)}$ ) consistent with both homogeneous and heterogeneous nucleation are observed and along with differing relationships between aerosol and  $N_{i(top)}$  depending on the prevailing meteorological situation and aerosol type. Away from the cloud top, the  $N_i$  displays a different sensitivity to these controlling factors, providing

10 a possible explanation to the low  $N_i$  sensitivity to temperature and INP observed in previous in-situ studies.

This satellite dataset provides a new way of investigating the response of cloud properties to meteorological and aerosol controls. The results presented in this work increase our confidence in the retrieved  $N_i$  and will form the basis for further study into the processes influencing ice and mixed phase clouds.

**1 Introduction**

- 15 Clouds play a central role in the Earth's energy budget, being responsible for large variations in the reflected shortwave and emitted longwave radiation (Stephens et al., 2012). The response of clouds to changing greenhouse gases and aerosols remains one of the largest uncertainties in understanding past and future climate changes (Boucher et al., 2013). Significant advances have been made into modelling and observing the role of aerosols in liquid clouds (e.g. Wang et al., 2011; Wood et al., 2011; Seifert et al., 2015; Gettelman, 2015; Ghan et al., 2016; Zuidema et al., 2016), especially through the use of retrievals of
- 20 the cloud droplet number concentration (e.g. Quaas et al., 2008; Gryspeerdt et al., 2016), but the impact of aerosols on high

clouds remains uncertain (Gettelman et al., 2012; Jensen et al., 2016; Zhou et al., 2016; Heyn et al., 2017). A large part of this uncertainty comes from the difficulty in retrieving cirrus cloud properties at a large enough scale to separate the roles of individual factors controlling the ice crystal number concentration ( $N_i$ ).

- A key microphysical property of ice clouds, the  $N_i$  links the aerosol environment to dynamic effects driving cloud updraughts and the generation of supersaturation (Pruppacher and Klett, 1997). Through changes in the ice crystal size, changes in the  $N_i$  can have far-reaching implications for a cloud, impacting the radiative (Liou, 1986; Fusina et al., 2007), precipitation and cloud lifetime properties (Lindsey and Fromm, 2008). The  $N_i$  is often used as a prognostic variable in two moment cloud microphysics schemes (e.g. Lohmann et al., 2007; Salzmann et al., 2010). This highlights a requirement to understand the controls on the  $N_i$  for improving our understanding and parametrisation of cloud processes. While aircraft measurements of
- 10 the  $N_i$  exist, they are restricted in space and time. They can be affected by shattering of ice crystals at the instrument inlet (McFarquhar et al., 2007; Jensen et al., 2009; Korolev et al., 2013) and difficulties in measuring the smallest crystals (O'Shea et al., 2016). In this paper, the new DARDAR-LIM satellite dataset described in part one (Sourdeval et al., submitted) allows the processes that control the  $N_i$  to be investigated globally.
- It is known that the temperature plays a strong role in determining the ice crystal nucleation rate. The homogeneous nucle-15 ation rate is a strong function of temperature and supersaturation (Koop et al., 2000), with atmospherically relevant nucleation 15 only taking place at temperatures colder than 235 K. This strong temperature dependence in the nucleation rate does not neces-17 sarily correspond to a strong temperature dependence in the Ni (Heymsfield and Miloshevich, 1993). A weak Ni temperature 18 dependence was found by Gayet et al. (2004). Krämer et al. (2009) found similar results, with a slight reduction in the Ni for 19 the coldest measurements. Higher Ni values have been observed at cold temperatures during ATTREX (Jensen et al., 2013a,
- 20 2016) than in Krämer et al. (2009), leading to a weak combined temperature dependence. However, using different datasets targeting different cloud types, Muhlbauer et al. (2014) and Jensen et al. (2013b) both showed an increase in  $N_i$  with decreasing temperature, demonstrating that there is still considerable uncertainty in the  $N_i$  temperature dependence.

The in-situ homogeneous nucleation of ice crystals is also dependent on the supersaturation (Koop et al., 2000; Lohmann and Kärcher, 2002), which is often generated through cooling due to vertical air motion. Large scale updraughts cannot reproduce observed cirrus properties on their own, the smaller scale variation in updraught provided by gravity waves is necessary (Kärcher and Ström, 2003) and is occasionally able to produce cirrus in regions of large scale subsidence (Muhlbauer et al., 2014). The scale data are the state of the scale of the scale data are the scale of the scale data are th

[revised manuscript text omitted]